# TTOpt: A Maximum Volume Quantized Tensor Train-based Optimization and its Application to Reinforcement Learning

**Konstantin Sozykin** [*][†]   **Andrei Chertkov** [*][†]   **Roman Schutski** [†]

**Anh-Huy Phan** [†]   **Andrzej Cichocki** [†][‡][§]   **Ivan Oseledets** [†][¶]

## Abstract

We present a novel procedure for optimization based on the combination of efficient quantized tensor train representation and a generalized maximum matrix volume principle. We demonstrate the applicability of the new Tensor Train Optimizer (TTOpt) method for various tasks, ranging from minimization of multidimensional functions to reinforcement learning. Our algorithm compares favorably to popular gradient-free methods and outperforms them by the number of function evaluations or execution time, often by a significant margin.

## 1   Introduction

In recent years learning-based algorithms achieved impressive results in various applications, ranging from image and text analysis and generation [52] to sequential decision making and control [40] and even quantum physics simulations [49]. The vital part of every learning-based algorithm is an optimization procedure, e.g., Stochastic Gradient Descent. In many situations, however, the problem-specific target function is not differentiable, too complex, or its gradients are not helpful due to the non-convex nature of the problem [30, 2, 58]. The examples include hyper-parameter selection during the training of neural models, policy optimization in reinforcement learning (RL), training neural networks with discrete (quantized) weights [57] or with non-differentiable loss functions [20]. In all these contexts, efficient direct gradient-free optimization procedures are highly needed.

Recently, [53] showed that an essential class of gradient-free methods, namely the evolutionary strategies (ES) [22, 26], are competitive in reinforcement learning problems. In RL, the goal is to find the agent's action distribution $\pi$ (the policy), maximizing some cumulative reward function $\mathsf{J}$. The policy is usually parameterized with a set of parameters $\boldsymbol{\theta}$. It follows that the reward is a function of the parameters of the policy: $\mathsf{J}(\pi(\boldsymbol{\theta})) = \mathsf{J}(\boldsymbol{\theta})$. The idea of [53] and similar works [36, 15, 10, 11] is to directly optimize the cumulative reward function $\mathsf{J}(\boldsymbol{\theta}) = \mathsf{J}(\theta_1, \theta_2, \ldots, \theta_d)$ with respect to the parameters of the policy. We pursued a similar approach to transform a traditional Markov Decision Process into an optimization problem (we provide the details in Appendix B.3). Although the agents trained with ES often demonstrate more rich behavior and better generalization compared to traditional gradient-based policy optimization, the convergence of ES is often slow [11].

---

[*]Equal Contribution, corresponding emails {konstantin.sozykin,a.chertkov}@skoltech.ru

[†]Center of Artificial Intelligence Technology, Skolkovo Institute of Science and Technology (Skoltech) Moscow, Russia

[‡]RIKEN Center for Advanced Intelligence Project (AIP), Tokyo, Japan

[§]Systems Research Institute, Polish Academy of Sciences, Warsaw, Poland

[¶]Artificial Intelligence Research Institute (AIRI), Moscow, Russia

36th Conference on Neural Information Processing Systems (NeurIPS 2022).

As an alternative to previous works, we present a tensor-based[1] gradient-free optimization approach and apply it to advanced continuous control RL benchmarks. The proposed algorithm, Tensor-Train (TT) Optimizer (**TTOpt**), works for multivariable functions with discrete parameters by reformulating the optimization problem in terms of tensor networks. Consider a function $\mathsf{J}(\boldsymbol{\theta}) : N^d \longrightarrow R$ of a $d$-dimensional argument $\boldsymbol{\theta}$, where each entry $\theta_k$ of the vector $\boldsymbol{\theta}$ takes a value in the discrete set $\{\omega_i\}_{i=1}^N$. The function $\mathsf{J}$ may be viewed as an implicitly defined $d$-dimensional tensor $\mathcal{J}$. Each entry in $\mathcal{J}$ is a value of $\mathsf{J}$ for some argument. Maximizing $\mathsf{J}$ is equivalent to finding the sets of indices $\{\boldsymbol{\theta}_{max}^{(m)}\}_{m=1}^M$ of maximal entries $\{j_{max}^{(m)}\}_{m=1}^M$ of $\mathcal{J}$.

By using only a tiny fraction of adaptively selected tensor elements (e.g., a small number of function evaluations), our method builds a representation of $\mathcal{J}$ in the TT-format and finds a set of the largest elements. Although the algorithm works only with functions of discrete arguments, the grid size for each parameter can be huge thanks to the efficiency of the TT-format. Fine discretization makes it possible to almost reach a continuous limit and obtain large precision with TTOpt. The strength of our approach is, however, the direct handling of discrete (quantized) parameters.

**Contributions.**   We propose an efficient gradient-free optimization algorithm for multivariable functions based on the low-rank TT-format and the generalized maximum matrix volume principle[2]. We demonstrate that our approach is competitive with a set of popular gradient-free methods for optimizing benchmark functions and neural network-based agents in RL problems. We empirically show that agents with discrete (heavily quantized) weights perform well in continuous control tasks. Our algorithm can directly train quantized neural networks, producing policies suitable for low-power devices.

## 2   Optimization with tensor train

In this section, we introduce a novel optimization algorithm. We show how to represent the optimization problems in the discrete domain efficiently and then formulate optimization as a sampling of the objective function guided by the maximum volume principle.

### 2.1   Discrete formulation of optimization problems

We first need to transfer the problem to the discrete domain to apply our method. It may seem that discretizing an optimization problem will make it harder. However, it will allow us to use powerful techniques for tensor network representation to motivate the algorithm. For each continuous parameter $\theta_k$ ($k = 1, 2, \ldots, d$) of the objective function $\mathsf{J}(\boldsymbol{\theta})$ we introduce a grid $\{\theta_k^{(n_k)}\}_{n_k=1}^{N_k}$. At each point $(\theta_1^{(n_1)}, \theta_2^{(n_2)}, \ldots, \theta_d^{(n_d)})$ of this grid with index $(n_1, n_2, \ldots, n_d)$ the objective function takes a value $\mathsf{J}(\theta_1^{(n_1)}, \theta_2^{(n_2)}, \ldots, \theta_d^{(n_d)}) \equiv \mathcal{J}[n_1, n_2, \ldots, n_d]$. We thus can regard the objective function as an implicit $d$-dimensional tensor $\mathcal{J}$ with sizes of the modes $N_1, N_2, \ldots, N_d$. Finding the maximum of the function $\mathsf{J}(\boldsymbol{\theta})$ translates into finding the maximal element of the tensor $\mathcal{J}$ in the discrete setting.

Notice that the number of elements of $\mathcal{J}$ equals: $|\mathcal{J}| = N_1 \cdot N_2 \cdot \ldots \cdot N_d \sim (\max_{1 \le k \le d} N_k)^d$. The size of $\mathcal{J}$ is exponential in the number of dimensions $d$. This tensor cannot be evaluated or stored for sufficiently large $d$. Fortunately, efficient approximations were developed to work with multidimensional arrays in recent years. Notable formats include Tensor Train (TT) [47, 46, 50, 44, 63], Tensor Chain/Tensor Ring [29, 66] and Hierarchical Tucker [21]. We use the most studied TT-format [13], but the extensions of our method to other tensor decompositions are possible.

---

[1]By tensors we mean multidimensional arrays with a number of dimensions $d$ ($d \ge 1$). A two-dimensional tensor ($d = 2$) is a matrix, and when $d = 1$ it is a vector. For scalars we use normal font, we denote vectors with bold letters and we use upper case calligraphic letters ($\mathcal{A}, \mathcal{B}, \mathcal{C}, \ldots$) for tensors with $d > 2$. Curly braces define sets. We highlight discrete and continuous scalar functions of multidimensional argument in the appropriate font, e.g., $\mathsf{J}(\cdot)$, in this case, the maximum and minimum values of the function are denoted by $J_{max}$ and $J_{min}$, respectively.

[2]We implemented the proposed algorithm within the framework of the publicly available software product: `https://github.com/AndreiChertkov/ttopt`.

## 2.2 Tensor Train decomposition

**Definition 2.1.** A tensor $\mathcal{J} \in \mathbb{R}^{N_1 \times N_2 \times \cdots \times N_d}$ is said to be in the TT-format [46] if its elements are represented by the following expression

$$\mathcal{J}[n_1, n_2, \ldots, n_d] = \sum_{r_0=1}^{R_0} \sum_{r_1=1}^{R_1} \cdots \sum_{r_d=1}^{R_d} \mathcal{G}_1[r_0, n_1, r_1]\mathcal{G}_2[r_1, n_2, r_2] \ldots \mathcal{G}_d[r_{d-1}, n_d, r_d], \quad (1)$$

where $n_k = 1, 2, \ldots, N_k$ for $k = 1, 2, \ldots, d$.

In TT-format the $d$-dimensional tensor $\mathcal{J}$ is approximated as a product of three-dimensional tensors $\mathcal{G}_k \in \mathbb{R}^{R_{k-1} \times N_k \times R_k}$, called TT-cores. The sizes of the internal indices $R_0, R_1, \cdots, R_d$ (with convention $R_0 = R_d = 1$) are known as TT-ranks. These ranks control the accuracy of the approximation.

The storage of the TT-cores, $\mathcal{G}_1, \mathcal{G}_2, \ldots, \mathcal{G}_d$, requires at most $d \cdot \max_{1 \le k \le d} N_k \cdot \left(\max_{0 \le k \le d} R_k\right)^2$ memory cells, and hence the TT-approximation is free from the curse of dimensionality[3] if the TT-ranks are bounded. The basic linear algebra operations (such as finding a norm, differentiation, integration, and others) can also be implemented in the TT-format with polynomial complexity in dimensionality and mode size.

**Building TT-approximation.** Several efficient schemes were proposed to find TT-approximation if all or some of the elements of the initial tensor are known or may be generated by the function's call. Examples include TT-SVD [47, 46], TT-ALS [27] and TT-CAM [48] (Cross Approximation Method in the TT-format).

We build upon the TT-CAM but modify it not to compute the approximation for the *entire* tensor, but rather to find a *small subset* of its maximal entries. The original algorithm builds a TT-approximation by adaptively requesting elements of the input tensor. As we will show below, these elements with high probability will have large absolute values. Based on this observation, we formulate a robust optimization algorithm for multivariate functions (either discrete or continuous). To simplify the understanding, we outline the approach for the two-dimensional case, and after that, we describe our gradient-free optimization method for the multidimensional case.

## 2.3 Maximal element in a matrix

The Cross Approximation Method (CAM) for matrices [18, 9, 1] is a well-established algorithm for building a rank-$R$ approximation $\tilde{J}$ of an implicitly given matrix $J$:

$$\boldsymbol{J} \simeq \tilde{\boldsymbol{J}}, \quad \tilde{\boldsymbol{J}} = \boldsymbol{J_C}\hat{\boldsymbol{J}}^{-1}\boldsymbol{J_R}, \quad (2)$$

where $\boldsymbol{J_C}$ consists of $R$ columns of $\boldsymbol{J}$, $\boldsymbol{J_R}$ is composed of $R$ rows of $\boldsymbol{J}$, and $\hat{\boldsymbol{J}}$ is a submatrix at their intersection. Such approximation (also called cross or skeleton decomposition) may be built iteratively using a well known alternating directions method and a maximum volume (maxvol) algorithm[4] [18], as we will sketch below.

**Intuition behind TTOpt.** The main interest in optimization problems is not the approximation (2) itself, but the following property of the resulting maximum volume submatrix $\hat{\boldsymbol{J}}$. [18] proved that if $\hat{\boldsymbol{J}}$ is an $R \times R$ submatrix of maximal volume (in selected rows and columns) then the maximal (by modulus) element $\hat{J}_{max} \in \hat{\boldsymbol{J}}$ bounds the absolute maximal element $J_{max}$ in the full matrix $\boldsymbol{J}$:

$$\hat{J}_{max} \cdot R^2 \ge J_{max}. \quad (3)$$

---

[3]The number of elements of an uncompressed tensor (hence, the memory required to store it) and the number of elementary operations required to perform computations with such a tensor grow exponentially in dimensionality. This problem is called the curse of dimensionality.

[4]The maxvol algorithm finds $R$ rows in an arbitrary non-degenerate matrix $\boldsymbol{A} \in \mathbb{R}^{N \times R}$ ($N > R$) which span a maximal-volume $R \times R$ submatrix $\hat{\boldsymbol{A}}$. The matrix $\hat{\boldsymbol{A}} \in \boldsymbol{A}$ has maximal value of the modulus of the determinant on the set of all nondegenerate square submatrices of the size $R \times R$. We describe the implementation of maxvol in Appendix A.1. The algorithm greedily rearranges rows of $\boldsymbol{A}$ to maximize submatrix volume. Its computational complexity is $O(NR^2 + KNR)$, where $K$ is a number of iterations.

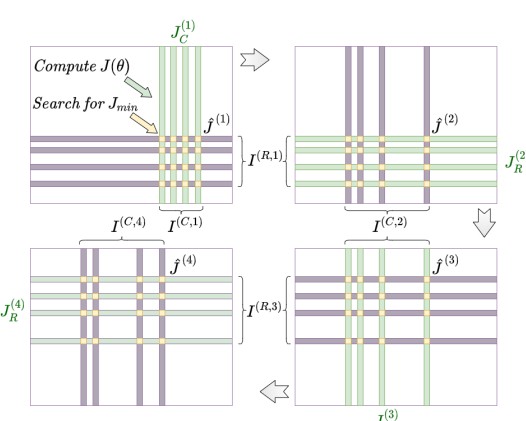

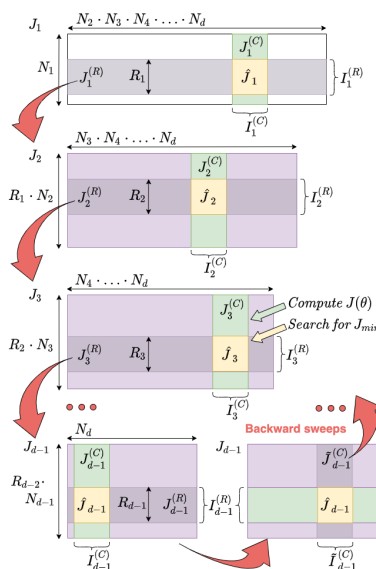

Figure 1: The scheme of the cross approximation algorithm for matrices using the alternating direction and maximal-volume principle. Green bars represent generated rows/columns; purple bars are rows/columns selected for generation in the next step by the maxvol algorithm. The method allows to find the optimum of the two dimensional function $\mathsf{J}(\theta_1, \theta_2)$.

Figure 2: Conceptual scheme of TTOpt algorithm based on the alternating direction and maximal-volume approaches for tensors. Only a small part of the tensor is explicitly generated during this procedure, as shown here with green columns. For the simplicity of presentation, the rows and columns selected at iterations are drawn as continuous blocks (they are not in practice).

This statement is evident for $R = 1$, and for the case $R > 1$ it gives an upper bound for the element. By using elementwise transformations of $\boldsymbol{J}$, this upper bound can be used to obtain a sequence that converges to the global optimum. The main idea of the maxvol-based methods is that it is easier to find a submatrix with a large volume rather than the element with the largest absolute value. Moreover, our numerical experiments show that this bound is pessimistic, and in practice, the maximal-volume submatrix contains the element which is very close to the optimal one.

**TTOpt algorithm for matrices.** The idea of the TTOpt algorithm for matrices is to iteratively search for the maximal volume submatrices in the column and row space of the implicitly given[5] input matrix $\boldsymbol{J} \in \mathbb{R}^{N_1 \times N_2}$. After $T$ iterations a series of "intersection" matrices $\{\hat{\boldsymbol{J}}^{(t)}\}_{t=1}^{T} \in \mathbb{R}^{R \times R}$ is produced. The maximal element is searched in these small submatrices. We schematically represent the TTOpt algorithm in Figure 1, and a description is given below:

1. At the initial stage, we set the expected rank of the approximation, $R$, and select $R$ random columns $I^{(C,1)}$. We then generate the corresponding column submatrix $\boldsymbol{J}_{\boldsymbol{C}}^{(1)} = \boldsymbol{J}[:, I^{(C,1)}] \in \mathbb{R}^{N_1 \times R}$. Using the maxvol algorithm, we find the maximal-volume submatrix $\hat{\boldsymbol{J}}^{(1)} \in \mathbb{R}^{R \times R}$ in $\boldsymbol{J}_{\boldsymbol{C}}^{(1)}$ and store its row indices in the list $I^{(R,1)}$.

2. The indices $I^{(R,1)}$ are used to generate a row submatrix $\boldsymbol{J}_{\boldsymbol{R}}^{(2)} = \boldsymbol{J}[I^{(R,1)}, :] \in \mathbb{R}^{R \times N_2}$. Then, using the maxvol algorithm, we find the maximal-volume submatrix $\hat{\boldsymbol{J}}^{(2)}$ in the matrix $\boldsymbol{J}_{\boldsymbol{R}}^{(2)}$ and store the corresponding column indices in the list $I^{(C,2)}$.

---

[5]The matrix is specified as a function $\mathsf{J}(\cdot)$ that allows to calculate the value of an arbitrary requested element $(n_1, n_2)$, where $1 \le n_1 \le N_1$ and $1 \le n_2 \le N_2$. We present the approach to approximate the value of the maximum modulus element of such a matrix. The method of finding the minimal or maximal elements within the framework of this algorithm will be described in Section 2.6.

3. We generate the related columns $\boldsymbol{J}_C^{(3)} = \boldsymbol{J}[:, I^{(C,2)}]$, apply again the maxvol algorithm to the column submatrix, and iterate the process until convergence.

4. The approximate value of the maximum modulus element of the matrix $\boldsymbol{J}$ is found as

$$\hat{J}_{max} = \max\left(\max(\hat{\boldsymbol{J}}^{(1)}), \max(\hat{\boldsymbol{J}}^{(2)}), \ldots, \max(\hat{\boldsymbol{J}}^{(T)})\right). \tag{4}$$

### 2.4 Optimization in the multidimensional case

As we explained previously, the target tensor, $\mathcal{J}$, is defined implicitly, e.g., by a multivariable function J. We propose a novel method to find the optimum in this implicit tensor. We outline the approach below and provide detailed algorithms in Appendix A.2.

As shown in Figure 2, we begin by considering the first unfolding[6] $\boldsymbol{J}_1 \in \mathbb{R}^{N_1 \times N_2 \ldots N_d}$ of the tensor $\mathcal{J}$ and select $R_1$ random columns $I_1^{(C)}$. Precisely, $I_1^{(C)}$ here is a list of $R_1$ random multi-indices of size $d - 1$, which specify positions along modes $k = 2$ to $k = d$. We then generate the submatrix $\boldsymbol{J}_1^{(C)} \in \mathbb{R}^{N_1 \times R_1}$ for all positions along the first mode (shown in green in Figure 2). Like in matrix case, we apply the maxvol algorithm to find the maximal-volume submatrix $\hat{\boldsymbol{J}}_1 \in \mathbb{R}^{R_1 \times R_1}$ and store the corresponding indices of $R_1$ rows in the list $I_1^{(R)}$.

In contrast with matrix case, we cannot generate the row submatrix $\boldsymbol{J}_1^{(R)} \in \mathbb{R}^{R_1 \times N_2 N_3 \ldots N_d}$ for the selected row indices $I_1^{(R)}$, since it contains an exponential number of elements. The following trick is used instead. We consider the implicit matrix $\boldsymbol{J}_1^{(R)}$ and reshape it to a new matrix $\boldsymbol{J}_2 \in \mathbb{R}^{R_1 N_2 \times N_3 \ldots N_d}$. We sample $R_2$ random columns $I_2^{(C)}$ in the matrix $\boldsymbol{J}_2$ and generate the entire small submatrix $\boldsymbol{J}_2^{(C)} \in \mathbb{R}^{R_1 N_2 \times R_2}$. Next we find the maximal-volume submatrix $\hat{\boldsymbol{J}}_2 \in \mathbb{R}^{R_2 \times R_2}$ in $\boldsymbol{J}_2^{(C)}$ and store the corresponding $R_2$ row multi-indices in the list $I_2^{(R)}$. The resulting submatrix $\boldsymbol{J}_2^{(R)} \in \mathbb{R}^{R_2 \times N_3 N_4 \ldots N_d}$ is then transformed (without explicitly evaluating its elements) into the matrix $\boldsymbol{J}_3 \in \mathbb{R}^{R_2 N_3 \times N_4 \ldots N_d}$.

We continue the described operations, called sweeps, until the last mode of the initial tensor, $\mathcal{J}$, is reached. After that, we repeat the process in the opposite direction, sampling now the row indices instead of the column indices. These sequences of forward and backward sweeps are continued until the algorithm converges to some row and column indices for all unfolding matrices[7] or until the user-specified limit on the number of requests to the objective function, J, is exceeded. Finally, after $T$ sweeps, the approximate value of the maximum modulo element can be found by the formula (4), as in the two-dimensional case.

Note that currently there is no analog of Eq. (3) in the multidimensional case, and hence there are no formal guarantees of the convergence of the sweeps to the global minimum, nor the rate of this convergence. The only guarantee is that the result will monotonically improve with iterations.

### 2.5 Complexity of the algorithm

It can be easily shown that the described algorithm requires to evaluate only $\mathcal{O}\left(d \cdot \max_{1 \leq k \leq d}\left(N_k R_k^2\right)\right)$ elements of the implicit tensor in one sweep. Thus, with a total number of sweeps $T$, we will have

$$\mathcal{O}\left(T \cdot d \cdot \max_{1 \leq k \leq d}\left(N_k R_k^2\right)\right), \tag{5}$$

calls to the objective function J. In practice, it turns out to be more convenient to limit the maximum number of function calls, $M$, according to the computational budget.

---

[6]The $k$-th unfolding $\boldsymbol{J}_k$ for the $d$-dimensional tensor $\mathcal{J} \in \mathbb{R}^{N_1 \times N_2 \times \cdots \times N_d}$ is the matrix $\boldsymbol{J}_k \in \mathbb{R}^{N_1 \ldots N_k \times N_{k+1} \ldots N_d}$, with elements $\boldsymbol{J}_k[\overline{n_1, \ldots, n_k}, \overline{n_{k+1}, \ldots, n_d}] \equiv \mathcal{J}[n_1, n_2, \ldots, n_d]$ for all indices.

[7]The TT-approximation (1) of the tensor $\mathcal{J}$ may be recovered from the generated columns $\boldsymbol{J}_k^{(C)}$ and maximal-volume submatrices $\hat{\boldsymbol{J}}_k$ ($k = 1, 2, \ldots, d$) as follows: $\boldsymbol{G}_k = \boldsymbol{J}_k^{(C)} \hat{\boldsymbol{J}}_k^{-1} \in \mathbb{R}^{R_{k-1} N_k \times R_k}$, where $\boldsymbol{G}_k$ is the 2-th unfolding of the $k$-th TT-core. However, we do not consider this point in more detail, since in our work, the main task is to find the minimum or maximum value of the tensor, not to construct its low-rank approximation.

If the time of a single call to $\mathsf{J}$ is significant, then the effort spent on the algorithm's operation will be negligible. Otherwise, the bottleneck will be the calculation of the maximal-volume submatrices by the $\mathsf{maxvol}$ algorithm. Taking into account the estimate of the $\mathsf{maxvol}$ complexity, given in Appendix A.1, it can be shown that in this case the complexity of our algorithm is

$$\mathcal{O}\left(T \cdot d \cdot \max_{1 \leq k \leq d}\left(N_k R_k^3\right)\right). \tag{6}$$

### 2.6 Implementation details

For the effective implementation of the TTOpt algorithm, the following important points should be taken into account (see also the detailed pseudocode in Appendix A.2).

**Stability.** Submatrices $\boldsymbol{J}_k^{(C)}$ (or $\boldsymbol{J}_C^{(k)}$ and $\boldsymbol{J}_R^{(k)}$ for the two-dimensional case) that arise during the iterations may degenerate, and in this case it is impossible to apply the $\mathsf{maxvol}$ algorithm. To solve this problem, we first calculate the QR decomposition for these matrices and then apply the $\mathsf{maxvol}$ to the corresponding $\boldsymbol{Q}$ factors[8].

**Rank selection.** We do not know in advance the exact ranks $R_1, R_2, \ldots, R_d$ of the unfolding matrices (or rank $R$ of the matrix $\boldsymbol{J}$ for the two-dimensional case). Therefore, instead of the $\mathsf{maxvol}$ algorithm, we use its modification, i.e., the $\mathsf{rect\_maxvol}$ algorithm[9] [39], within the framework of which several ("most important") rows are added to the maximal-volume submatrix. In this case, we have $\hat{\boldsymbol{J}}_1^{(C)} \in \mathbb{R}^{(R_1+\Delta R_1) \times R_1}$, $\hat{\boldsymbol{J}}_2^{(C)} \in \mathbb{R}^{(R_2+\Delta R_2) \times R_2}$, etc. Note that the final approximation obtained in this case may have an overestimated rank, which, if necessary, can be reduced by appropriate rounding, for example, by truncated SVD decomposition.

**Mapping function.** Maximal-volume submatrices contain the maximum modulus element but not the minimum or maximum element of the tensor (i.e., the sign is not taken into account). We introduce a dynamic mapping function to find the global minimum (or maximum). Instead of $\mathsf{J}$ at each step of the algorithm, we evaluate[10]:

$$\mathsf{g}(\boldsymbol{x}) = \frac{\pi}{2} - \mathsf{atan}\left(\mathsf{J}(\boldsymbol{x}) - J_{min}\right), \tag{7}$$

where $J_{min}$ is the current best approximation for the minimum element of the tensor.

**Quantization.** To reach high accuracy, we often need fine grids. In the case when the sizes of the tensor modes $N_1, N_2, \ldots, N_d$ are large, the sizes of unfolding matrices become large, which leads to a significant increase in computational complexity of the $\mathsf{maxvol}$ algorithm. To solve this problem, we apply additional compression based on quantization of the tensor modes [45]. Assume without the loss of generality that the size of each mode is $N_k = P^q$ ($k = 1, 2, \ldots, d$; $P \geq 2$; $q \geq 2$). Then we can reshape the original $d$-dimensional tensor $\mathcal{J} \in \mathbb{R}^{N_1 \times N_2 \times \cdots \times N_d}$ into the tensor $\tilde{\mathcal{J}} \in \mathbb{R}^{P \times P \times \cdots \times P}$ of a higher dimension $d \cdot q$, but with smaller modes of size $P$. The TTOpt algorithm can be applied for this "long" tensor instead of the original one[11]. We found that this idea significantly boosts the accuracy of our algorithm and reduces the complexity and execution time. Typically, $P$ is taken as small as possible, e.g., $P = 2$.

## 3 Experiments

To demonstrate the advantage of the proposed optimization method, we tested TTOpt on several numerical problems. First, we consider analytical benchmark functions and then the practically

---

[8]It can be shown that this operation does not increase the complexity estimate (6) of the algorithm.

[9]The $\mathsf{rect\_maxvol}$ algorithm allows to find $R + \Delta R$ rows in an arbitrary nondegenerate matrix $\boldsymbol{A} \in \mathbb{R}^{N \times R}$ ($N > R$, $N \geq R + \Delta R$) which form an approximation to the *rectangular* maximal-volume submatrix of the given matrix $\boldsymbol{A}$. Details about $\mathsf{rect\_maxvol}$ are provided in Appendix A.1.

[10]The mapping function should be continuous, smooth, and strictly monotone. There are various ways to choose such a function. However, during test runs, it turned out that the proposed function (7) is most suitable.

[11]If the maximal rank $R > P$, we select all indices for first $k$ modes until $P^k > R$. We note however that this detail does not change the global behavior of the TTOpt algorithm.

Table 1: Comparison of the TTOpt optimizer versus baselines in terms of the final error $\epsilon$ (absolute deviation of the obtained optimal value relative to the global minimum) and computation time $\tau$ (in seconds) for various benchmark functions. See Table 1 in Appendix B.1 with the list of functions and their properties. The reported values are averaged over 10 independent runs. The upper half of the table presents gradient free (zeroth order) methods, and the lower half is for first and second order methods.

| | | F1 | F2 | F3 | F4 | F5 | F6 | F7 | F8 | F9 | F10 |
|---|---|---|---|---|---|---|---|---|---|---|---|
| TTOPT | $\epsilon$ | **3.9E-06** | **2.9E-07** | 1.8E-12 | 4.4E-15 | 2.8E-02 | **1.1E-01** | 5.5E-09 | **4.6E-11** | 1.8E-01 | **1.3E-04** |
| | $\tau$ | 2.61 | 2.44 | 2.45 | 2.52 | 2.40 | 2.48 | 2.39 | 2.60 | 2.32 | 2.44 |
| GA | $\epsilon$ | 9.7E-02 | 8.4E-03 | 5.8E-03 | 2.0E+00 | **3.9E-04** | 1.0E+01 | 1.2E-01 | 7.9E-01 | **6.2E-03** | 4.2E+03 |
| | $\tau$ | 6.21 | 4.56 | 5.09 | 4.87 | 5.85 | 5.69 | 5.05 | 5.04 | 5.04 | 4.70 |
| OPENES | $\epsilon$ | 1.8E-01 | 1.2E-02 | 1.7E-02 | 2.0E+00 | 1.2E-03 | 9.7E+00 | 3.8E+00 | 2.1E+00 | 1.8E-02 | 4.2E+03 |
| | $\tau$ | 2.62 | 1.08 | 1.62 | 1.08 | 2.41 | 2.04 | 1.30 | 1.39 | 1.62 | 1.12 |
| CMAES | $\epsilon$ | 5.1E+287 | 3.1E-01 | **9.3E-77** | 2.0E+00 | 7.6E+289 | 1.9E+01 | 5.5E-02 | 9.3E+01 | 5.3E+289 | 1.7E+282 |
| | $\tau$ | 10.36 | 8.50 | 9.40 | 9.13 | 12.86 | 9.76 | 9.04 | 9.13 | 11.15 | 8.86 |
| DE | $\epsilon$ | 1.1E+00 | 4.3E-02 | 3.3E-02 | 9.0E-05 | 1.8E-01 | 2.6E-01 | 2.0E-01 | 6.2E+00 | 6.6E-01 | 3.8E+02 |
| | $\tau$ | 38.91 | 38.06 | 51.05 | 39.48 | 41.35 | 41.40 | 41.34 | 41.31 | 37.97 | 38.64 |
| NB | $\epsilon$ | 1.5E+01 | 6.5E+00 | 3.9E+01 | 1.2E-01 | 2.4E+01 | 6.6E+00 | 2.6E+10 | 6.3E+01 | 3.4E+00 | 3.2E+03 |
| | $\tau$ | 45.23 | 46.98 | 37.50 | 45.91 | 48.03 | 37.16 | 40.05 | 44.95 | 44.06 | 46.91 |
| PSO | $\epsilon$ | 1.2E+01 | 5.3E+00 | 3.5E+01 | 9.8E-02 | 2.0E+01 | 2.5E-01 | 2.0E+10 | 2.3E+01 | 5.1E-01 | 2.9E+03 |
| | $\tau$ | 47.19 | 47.04 | 45.50 | 43.39 | 46.80 | 44.97 | 46.46 | 42.78 | 43.15 | 47.13 |
| BFGS | $\epsilon$ | 1.9E+01 | 2.1E+00 | 1.9E+01 | 4.3E-13 | 1.2E-02 | 6.4E+00 | 2.4E-05 | 7.0E+01 | 4.2E+00 | 2.1E+03 |
| | $\tau$ | 0.01 | 0.02 | 0.03 | 0.00 | 0.02 | 0.01 | 0.04 | 0.01 | 0.01 | 0.00 |
| L-BFGS | $\epsilon$ | 1.9E+01 | 1.9E+00 | 4.0E-10 | 4.3E-13 | 1.2E-02 | 4.5E+00 | 4.5E-10 | 7.0E+01 | 4.2E+00 | 2.1E+03 |
| | $\tau$ | 0.01 | 0.06 | 0.05 | 0.00 | 0.01 | 0.02 | 0.02 | 0.01 | 0.01 | 0.01 |
| CG | $\epsilon$ | 1.9E+01 | 3.4E+00 | N/A | **0.0E+00** | 2.0E-02 | 4.4E+00 | 2.9E-12 | 7.0E+01 | 4.2E+00 | 2.1E+03 |
| | $\tau$ | 0.01 | 0.07 | N/A | 0.00 | 0.01 | 0.05 | 0.02 | 0.01 | 0.03 | 0.00 |
| NCG | $\epsilon$ | 1.9E+01 | 3.4E+00 | 1.7E-19 | **0.0E+00** | 7.4E-02 | 6.4E+00 | 2.1E-12 | 7.0E+01 | 4.2E+00 | 2.2E+03 |
| | $\tau$ | 0.01 | 0.06 | 0.04 | 0.00 | 0.01 | 0.06 | 0.01 | 0.00 | 0.02 | 0.01 |
| NEWTON | $\epsilon$ | 1.9E+01 | 2.2E+00 | 1.1E+11 | **0.0E+00** | 3.2E-02 | 6.4E+00 | **2.8E-24** | 7.0E+01 | 4.2E+00 | 2.1E+03 |
| | $\tau$ | 0.01 | 0.02 | 0.01 | 0.00 | 0.01 | 0.10 | 0.01 | 0.01 | 0.01 | 0.00 |
| TR NCG | $\epsilon$ | 1.9E+01 | 6.5E+00 | 2.4E-10 | 4.0E-12 | 4.8E+00 | 4.4E+00 | 1.2E-14 | 7.0E+01 | 4.2E+00 | 2.1E+03 |
| | $\tau$ | 0.01 | 0.12 | 0.04 | 0.00 | 0.02 | 0.03 | 0.01 | 0.00 | 0.02 | 0.01 |
| TR | $\epsilon$ | 1.9E+01 | 7.0E+00 | 3.2E-13 | 4.0E-12 | 6.1E+01 | 2.6E+00 | 5.3E-11 | 7.0E+01 | 4.2E+00 | 2.1E+03 |
| | $\tau$ | 0.02 | 12.97 | 0.06 | 0.00 | 0.06 | 0.05 | 0.02 | 0.01 | 0.01 | 0.01 |

significant problem of optimizing the parameters of the RL agent. We select GA (Genetic Algorithm [26, 60]), openES (basic version of OpenAI Evolution Strategies [53]) and cmaES (the Covariance Matrix Adaptation Evolution Strategy [22]) as baselines for both experiments. Additionally we used DE (Differential Evolution [59]), NB (NoisyBandit method from Nevergrad [5]) and PSO (Particle Swarm Optimization [24, 33]) for benchmark functions[12]. We also compared the proposed approach with gradient-based methods applied for all benchmark functions[13]: BFGS (Broyden–Fletcher–Goldfarb–Shanno algorithm), L-BFGS (Limited-memory BFGS), CG (Conjugate Gradient algorithm), NCG (Newton CG algorithm), Newton (Newton Exact algorithm), TR NCG (Trust-Region NCG algorithm) and TR (Trust-Region Exact algorithm).

According to our approach, the TTOpt solver has the following configurable parameters: **a** and **b** are lower and upper grid bounds (for simplicity, we use the same value for all dimensions); **R** is a rank (for simplicity, we use the same value for all unfolding matrices); **P** is a submode size (mode size of the quantized tensor; for simplicity, we use the same value for all dimensions); **q** is the number of submodes in the quantized tensor (each mode of the original tensor has size $N = P^q$); **M** is a limit on the number of requests to the objective function.

### 3.1 Benchmark functions minimization

To analyze the effectiveness of the TTOpt, we applied it to 10-dimensional benchmark functions with known global minimums; see Table 1 in Appendix B.1 with the list of functions and their properties (note that some of the considered benchmarks are multimodal non-separable functions). Also, in Appendix B.1, we present a more detailed study of the TTOpt solver and the dependence of the accuracy on the value of its parameters ($R$, $q$ and $M$).

---

[12]We used implementations of the methods from available packages estool (`https://github.com/hardmaru/estool`), pycma (`https://github.com/CMA-ES/pycma`, and nevergrad (`https://github.com/facebookresearch/nevergrad`).

[13]We used implementations from the package `https://github.com/rfeinman/pytorch-minimize`). We carried out computations with all methods from this library, except for Trust-Region GLTR (Krylov) and Dogleg methods, for which the calculation ended with an error for most benchmarks.

Table 2: The result of the TTOpt optimizer in terms of the final error $\epsilon$ (absolute deviation of the obtained optimal value relative to the global minimum) and computation time $\tau$ (in seconds) for benchmark *F1* (Ackley function) for various dimension numbers.

| DIMENSION | $d = 10$ | $d = 50$ | $d = 100$ | $d = 500$ |
|---|---|---|---|---|
| ERROR, $\epsilon$ | 3.9E-06 | 3.9E-06 | 3.9E-06 | 3.9E-06 |
| TIME, $\tau$ | *3.1* | *40.1* | *143.9* | *3385.3* |

Table 3: Mean $\mathbb{E}$ and standard deviation $\sigma$ of the final cumulative reward. The environments are encoded using capital letters **S**wimmer-v3, **L**unarLanderContinuous-v2, **I**nvertedPendulum-v2 and **H**alfCheetah-v3. The left sub-table is for mode size $N = 3$, another one is for mode size $N = 2^8$. Results are averaged over 10 random seeds.

| | | $S(3)$ | $L(3)$ | $I(3)$ | $H(3)$ | $S(2^8)$ | $L(2^8)$ | $I(2^8)$ | $H(2^8)$ |
|---|---|---|---|---|---|---|---|---|---|
| TTOPT | $\mathbb{E}$ | **357.50** | **290.29** | **1000.00** | **4211.02** | 311.82 | **286.87** | **1000.00** | 2935.90 |
| | $\sigma$ | **6.59** | **24.40** | **0.00** | 211.94 | 29.61 | **21.65** | **0.00** | 544.11 |
| GA | $\mathbb{E}$ | 349.91 | 283.05 | 893.00 | 2495.37 | **359.79** | 213.75 | 222.86 | **3085.80** |
| | $\sigma$ | 10.04 | 16.28 | 283.10 | 185.11 | **4.21** | 99.67 | 342.79 | **842.76** |
| CMAES | $\mathbb{E}$ | 342.31 | 214.55 | 721.00 | 2549.83 | 340.54 | 221.95 | 621.00 | 2879.46 |
| | $\sigma$ | 36.07 | 93.79 | 335.37 | 501.08 | 78.90 | 133.80 | 472.81 | 929.55 |
| OPENES | $\mathbb{E}$ | 318.39 | 114.97 | 651.86 | 2423.16 | 109.39 | 73.08 | 224.71 | 1691.22 |
| | $\sigma$ | 44.61 | 113.48 | 436.37 | 602.43 | 40.11 | 163.33 | 217.51 | 976.96 |

In all experiments with baselines (GA, openES, cmaES, DE, NB, PSO, BFGS, L-BFGS, CG, NCG, Newton, TR NCG and TR), we used default parameter values. In Appendix B.2 we also present the additional experiments with Bayesian Optimization [37]. For TTOpt we selected rank[14] $R = 4$, submode size $P = 2$ and the number of submodes $q = 25$. For all methods, a limit on the number of requests to the objective function is chosen as $M = 10^5$. All calculations are performed on a standard laptop.

The results are demonstrated in Table 1. For each method we list the absolute deviation of the result $\hat{J}_{min}$ from the global minimum $J_{min}$, i.e., $\epsilon = |\hat{J}_{min} - J_{min}|$. We also present the total running time, $\tau$. Compared to other benchmarks, TTOpt is consistently fast, accurate and avoids random failures to converge seen in other algorithms. Additionally, TTOpt turns out one of the fastest gradient-free algorithms (GA, openES, cmaES, DE, NB, PSO), despite a simple Python implementation.

One of the advantages of the proposed TTOpt approach is the possibility of its application to essentially multidimensional functions. In Table 2 we present the result of TTOpt for the *F1* benchmark function of various dimensionality (results for other benchmarks are in Appendix B.1). Note that as a limit on the number of requests to the objective function we chose $M = 10^4 \cdot d$, and the values of the remaining parameters were chosen the same as above. As can be seen, even for $500$-dimensional functions, the TTOpt method gives a fairly accurate result.

## 3.2 Application of TTOpt to Reinforcement Learning

We used several continuous RL tasks implemented in Mujoco [61] and OpenAI-GYM [8]: Swimmer-v3 [16], LunarLanderContinuous-v2, InvertedPendulum-v2 and HalfCheetah-v3 [64]. In all experiments, the policy $\pi$ is represented by a neural network with three hidden layers and with tanh and ReLU activations. Each layer is a convolution layer. See additional details about hyperparameters in Table 6 in Appendix B.

We discretize (quantize) the values of agent's weights. The TTOpt method is used to optimize discrete agent's weights in order to maximize the cumulative reward of the episode. This corresponds to on-policy learning.

---

[14]We chose rank $R$ using the following heuristic. The minimal number of sweeps is fixed as $T = 5$. It follows that the algorithm will need $2 \cdot T \cdot (dq) \cdot P \cdot R^2$ function calls. With a given limit on the number of function requests $M$, the rank can be estimated as $R \leq \sqrt{\frac{M}{2 \cdot T \cdot d \cdot q \cdot P}}$.

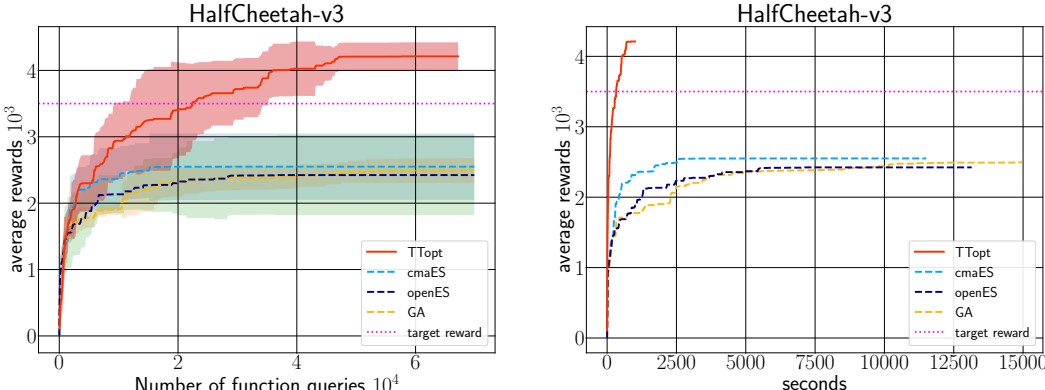

Figure 3: Training curves of TTOpt and baselines for HalfCheetah-v3 ($N = 3$). The upper plot is the average cumulative reward versus the number of interactions with the environment. The lower plot is the same versus execution time. The reward is averaged for seven seeds. The shaded area shows the difference of one standard deviation around the mean. See similar plots for other environments in Appendix B.5.

To properly compare TTOpt with other methods, we propose modified evolutionary baselines that enforce constrained parameter domain. We adapt penalty term and projection techniques from [31, 6] to introduce constraints, see Appendix B.4 for details.

First, we run benchmarks with small mode size $N = 3$ with lower and upper grid bounds $\pm 1$. Another series of experiments was done with finer mode of size $N = 2^q$ with the same bounds. These experiments model the case of neural networks with discrete (quantized) weights which use $q$-bits quantization. Finally, we provide the results of using TTOpt as a fine-tuning procedure for linear policies from [36].

We present characteristic training curves based on the number of environment interactions and execution time for the HalfCheetah-v3 experiment ($N = 3$) in Figure 3. Training curves for other environments can be found in Appendix B.5, in Figure 3 for $N = 3$ and in Figure 4 for $N = 256$. TTOpt consistently outperforms all other baselines on the coarse grid with mode size $N = 3$. Our method is still best for finer grids with mode size $N = 256$ on InvertedPendulum-v2 and LunarLanderContinuous-v2, and second-best on HalfCheetah-v3. Moreover, TTOpt has significantly lower execution time compared to evolutionary baselines. Another interesting observation is that the training curves of TTOpt have low dispersion, e.g., the algorithm performs more consistently than the baselines (see Appendix B.5). Table 3 summarizes the experiments for the coarse and fine grids. Results for fine-tuning of linear policies are presented in Table 5 in Appendix B.5. We also did rank and reward dependency study in Appendix B.

## 4   Related work

In the case of high dimensional optimization, evolutionary strategies (ES) [56, 42] are one of the most advanced methods of black-box optimization. This approach aims to optimize the parameters of the search distribution, typically a multivariate Gaussian distribution, to maximize the objective function. Finite difference schemes are commonly used to approximate gradients of the parameters of the search distribution. Numerous works proposed techniques to improve the convergence of ES [42]. [65] proposed to use second-order natural gradient of [12] to generate updates, while [22] suggested to include the history of recent updates to generate next ones. [35] presented the concept of surrogate gradients for faster convergence.

Another series of works aimed to reduce the high sampling complexity of ES. In [11] the authors described how to use active subspaces [14] to reduce the number of objective function calls dynamically.

Plenty of the already mentioned works in gradient-free optimization specifically applied these methods to RL tasks [53, 10, 11, 36, 15, 23]. Overall, the performance of ES-based methods is comparable

to conventional policy gradients, especially if the number of model parameters is small [54, 55]. Another advantage of ES over policy gradient is that it produces more robust and diverse policies [32] by eliminating the problem of delayed rewards and short length time horizons. Finally, evolutionary approaches are suitable for the problems with non-Markovian properties [23].

Other metaheuristic [25, 38] and classical optimization [51] techniques are also studied within RL scope. The examples include simulated annealing [4], particle swarm optimization [24, 33] and even classical Nelder-Mead algorithm [41, 43]. These methods, however, are not tested on common RL task sets. Several other works combined evolutionary methods with RL to achieve better performance in complex scenarios [28, 17], e.g., AlphaStar [62, 3].

Finally, low-rank tensor approximations have been applied to RL problems in settings different from ours, including multi-agent scenarios [34], and improving dynamic programming approaches [19, 7]. The idea of our method is quite different from the presented works, especially in the RL area.

## 5    Conclusion

We proposed a new discrete optimization method based on quantized tensor-train representation and maximal volume principle. We demonstrate its performance for analytical benchmark functions and reinforcement learning problems. Our algorithm is more efficient under a fixed computational budget than baselines, especially on discrete domains. Moreover, the execution time for TTOpt is lower by a significant margin compared with other baselines. Finally, we show that the agents with discrete parameters in RL can be as efficient as their continuous parameter versions. This observation supports the broad adoption of quantization in machine learning. We hope that our approach will serve as a bridge between continuous and discrete optimization methods.

## Acknowledgments and Disclosure of Funding

The work was supported by Ministry of Science and Higher Education grant No. 075-10-2021-068.

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
