# TTOpt: A Maximum Volume Quantized Tensor Train-based Optimization and its Application to Reinforcement Learning. Supplementary Material

**Konstantin Sozykin** [* †]  **Andrei Chertkov** [* †]  **Roman Schutski** [†]

**Anh-Huy Phan** [†]  **Andrzej Cichocki** [† ‡ §]  **Ivan Oseledets** [† ¶]

## A  Description of MaxVol and TTOpt algorithms

### A.1  Maxvol Algorithm

The search for the maximal element in a large matrix can be significantly simplified if one can obtain a "good" submatrix, for example, a maximal-volume submatrix. Finding the maximal-volume submatrix of a nondegenerate matrix $A \in \mathbb{R}^{N \times R}, N > R$ is an NP-hard problem. In this paper, we adapted the maxvol algorithm [3] which can find a submatrix $C \in \mathbb{R}^{R \times R}$ of $A$, such that its determinant is close to maximum in absolute value. The algorithm selects a set of $R$ rows denoted by $I \subset [1, \dots, N]$ which form the matrix $C = A[I, :]$.

For the given tolerance threshold $\epsilon$ ($\epsilon \geq 1$ and close to one), we find the set $I$ as follows:

1. Compute the LU-decomposition $A = PLU$ and store the permutation of the first $R$ rows (according to the matrix $P$) in the list $I$.
2. Generate the matrix $Q \in \mathbb{R}^{R \times N}$ as a solution to the linear system with an upper triangular matrix $U$: $U^T Q = A^T$.
3. Compute the matrix $B \in \mathbb{R}^{N \times R}$ as a solution to the linear system with the lower triangular matrix $L$: $(L[: R, :])^T B^T = Q$.
4. Find the maximum modulo element $b = B_{i,j}$ of the matrix $B$. If $|b| \leq \epsilon$, then terminate the algorithm[1] by returning the current list $I$ and matrix $B$.
5. Update the matrix $B \leftarrow B - B[:, j] (B[i, :] - e_j^T) b^{-1}$, where $e_j$ is the $j$-th unit basis vector.
6. Update the list $I$ as $I[j] \leftarrow i$.
7. Return to step 4.

It can be shown that at each step $k$ of the algorithm the volume of the submatrix $C$ increases by a factor not less than $\epsilon$. Therefore, the estimate for the number of iterations, $K$, is the following:

$$K = \frac{\log\left(\,|\det(\hat{C})|\,\right) - \log\left(\,|\det(C^{(0)})|\,\right)}{\log \epsilon}, \tag{1}$$

---

[*]Equal Contribution, corresponding emails {konstantin.sozykin,a.chertkov}@skoltech.ru

[†]Center of Artificial Intelligence Technology, Skolkovo Institute of Science and Technology (Skoltech) Moscow, Russia

[‡]RIKEN Center for Advanced Intelligence Project (AIP), Tokyo, Japan

[§]Systems Research Institute, Polish Academy of Sciences, Warsaw, Poland

[¶]Artificial Intelligence Research Institute (AIRI), Moscow, Russia

[1]In addition to the indices of the rows $I$ that form the maximal-volume submatrix $C \in \mathbb{R}^{R \times R}$, we also obtain the matrix of coefficients $B \in \mathbb{R}^{N \times R}$ such that $A = BC$.

where $\hat{C}$ is the exact maximal-volume submatrix and $C^{(0)}$ is an initial approximation obtained from the LU-decomposition (see step 1 above). The computational complexity of the algorithm is the sum of the initialization complexity and the complexities of $K$ iterations. The complexity equals

$$\mathcal{O}\left(NR^2 + KNR\right). \tag{2}$$

In the context of the maximal matrix element search problem, other definitions of "good" submatrices are possible. For example, one can search for a rectangular submatrix containing rows or columns which span the largest volume. The rect_maxvol algorithm can be used in this case. We employ rect_maxvol [7] to choose rectangular submatrices within the framework of the TTOpt algorithm to adaptively increase the rank.

Consider an arbitrary nondegenerate matrix $A \in \mathbb{R}^{N \times R}$ ($N > R$). We search for a rectangular submatrix $\hat{C} \in \mathbb{R}^{(R+\Delta R) \times R}$, $R + \Delta R < N$, which maximizes the rectangular volume $\sqrt{\det\left(\hat{C}^T \hat{C}\right)}$ objective. An approximation $C$ to $\hat{C}$ can be found as follows. The first $R$ rows of $C$ are obtained by the maxvol algorithm. The following rect_maxvol algorithm will find additional $\Delta R$ rows:

1. First, generate $R$ indices of rows $I \in \mathbb{N}^R$ and the coefficient matrix $B \in \mathbb{R}^{N \times R}$ of the initial approximation using the maxvol algorithm.

2. Compute the vector $l \in \mathbb{R}^N$ containing the norms of the rows: $l[j] = (B[j,:])^T B[j,:]$ for $j = 1, 2, \ldots, N$.

3. Find the maximum modulo element of the vector $l$, i.e. $i = argmax(l)$.

4. If the current length $R + \Delta R$ of the vector $I$ is greater than or equal to $R^{(max)}$ or if $l[i] \le \tau^2$, then terminate the algorithm[2] by returning the current list $I$ and matrix $B$.

5. Update the coefficient matrix as

$$B := \left[ B - \frac{B(B[i,:])^T B[i,:]}{1 + B[i,:](B[i,:])^T} \quad \frac{B(B[i,:])^T}{1 + B[i,:](B[i,:])^T} \right].$$

6. Update the vector of row norms

$$l[j] := l[j] - \frac{|B[j,:](B[i,:])^T|^2}{1 + B[i,:](B[i,:])^T}, \quad j = 1, 2, \ldots, N.$$

7. Add current row index $i$ to the list $I$.

8. Return to step 3.

The approximate computational complexity of this algorithm according to the work [7] is $O(NR^2)$, and the expected number of rows in the resulting submatrix is $2R - 1$ for the case $\tau = 1$.

## A.2   TTOpt Algorithm

In Algorithm A.1 we present the details of the TTOpt implementation. The TTOpt algorithm builds the TT proxy of the minimized function which is iteratively updated. The procedure of updating the TT proxy is outlined in Algorithm A.2 (function update_left, that updates core tensors of the network from right to left ) and in Algorithm A.3 (function update_right, updates core tensors of the network from left to right ). The requests to the objective function and the transformation of resulting values are presented in Algorithm A.4 (function eval).

The procedure begins by building one-dimensional uniform grids $x_i$ ($i = 1, 2, \ldots, d$) for the function argument along each mode, using the specified bounds of the rectangular search domain $\Omega$. Note that, if necessary, arbitrary nonuniform grids can be used, taking into account the specific features of the function under consideration.

---

[2]As a criterion for stopping the algorithm, we consider either the achievement of the maximum number of rows in the maximal-volume submatrix ($R^{(max)}$), or the sufficiently small norm of the remaining rows in the matrix of coefficients $B$.

**Algorithm A.1:** Multivariable function minimizer TTOpt.

---

**Data:** function $J(\boldsymbol{\theta})$, where $\boldsymbol{\theta} \in \Omega \subset \mathbb{R}^d$; boundary points of the rectangular domain $\Omega = [a_1, b_1] \times [a_2, b_2] \times \cdots \times [a_d, b_d]$; the number of grid points for every dimension $N_1, N_2, \ldots, N_d$; number of inner iterations (sweeps) $k_{max}$; maximum TT-rank $r_{max}$.

**Result:** approximation of the spatial point $\boldsymbol{\theta}_{min} \in \mathbb{R}^d$ at which the function $J$ reaches its minimum in the region $\Omega$ and a corresponding function value $J_{min} \in \mathbb{R}$.

1   // Construct a uniform grid $\boldsymbol{x}_1, \boldsymbol{x}_2, \ldots, \boldsymbol{x}_d$:
2   Set $\boldsymbol{x}_i[m] = a_i + (b_i - a_i) \cdot \frac{m-1}{N_i - 1}$
3   for all $i = 1, \ldots, d$ and $m = 1, 2, \ldots, N_i$.
4   // Initialize the TT-ranks $R_0, R_1, \ldots, R_d$:
5   Set $R_0 = 1$.
6   **for** $i = 1$ **to** $(d-1)$ **do**
7     |   Set $R_i = \min(R_{i-1} \cdot N_i, R_i \cdot N_i, r_{max})$.
8   **end**
9   Set $R_d = 1$.
10   // Initialize set of points of interest $\boldsymbol{X}_0, \boldsymbol{X}_1, \ldots, \boldsymbol{X}_d$:
11   Set $\boldsymbol{X}_0 = None$.
12   **for** $i = 0$ **to** $(d-2)$ **do**
13     |   Set $\boldsymbol{G}_i = \mathsf{random}(R_i \cdot N_i, R_{i+1})$ from the standard normal distribution.
14     |   Compute QR-decomposition $\boldsymbol{Q}, \boldsymbol{R} = \mathsf{QR}(\boldsymbol{G}_i)$.
15     |   Compute indices $\boldsymbol{m}_{opt} = \mathsf{maxvol}(\boldsymbol{Q})$.
16     |   Set $\boldsymbol{X}_{i+1} = \mathsf{update\_right}(\boldsymbol{X}_i, \boldsymbol{x}_i, N_i, R_i, \boldsymbol{m}_{opt})$.
17   **end**
18   Set $\boldsymbol{X}_d = None$.
19   // Iterate in a loop to find optimal $\boldsymbol{\theta}_{min}$ and $J_{min}$:
20   Set $\boldsymbol{\theta}_{min} = None$ and $J_{min} = +\infty$.
21   **for** $k = 1$ **to** $k_{max}$ **do**
22     |   // Traverse the TT-cores from right to left:
23     |   **for** $i = d$ **to** $1$ **do**
24     |     |   Compute $\boldsymbol{z}, \boldsymbol{\theta}_{min}, J_{min} = \mathsf{eval}(*)$.
25     |     |   Reshape $\boldsymbol{z}$ to matrix $\boldsymbol{Z} \in \mathbb{R}^{R_i \times n_i \cdot R_{i+1}}$.
26     |     |   Compute $\boldsymbol{Q}, \boldsymbol{R} = \mathsf{QR}(\boldsymbol{Z}^T)$.
27     |     |   Compute indices $\boldsymbol{m}_{opt} = \mathsf{rect\_maxvol}(\boldsymbol{Q}[:, 0 : R[i]])$.
28     |     |   Set $\boldsymbol{X}_i = \mathsf{update\_left}(\boldsymbol{X}_{i+1}, \boldsymbol{x}_i, N_i, R_{i+1}, \boldsymbol{m}_{opt})$.
29     |   **end**
30     |   // Traverse the TT-cores from left to right:
31     |   **for** $i = 1$ **to** $d$ **do**
32     |     |   Compute $\boldsymbol{z}, \boldsymbol{\theta}_{min}, J_{min} = \mathsf{eval}(*)$.
33     |     |   Reshape $\boldsymbol{z}$ to matrix $\boldsymbol{z} \in \mathbb{R}^{R_i \cdot n_i \times R_{i+1}}$.
34     |     |   Compute $\boldsymbol{Q}, \boldsymbol{R} = \mathsf{QR}(\boldsymbol{Z})$.
35     |     |   Compute indices $\boldsymbol{m}_{opt} = \mathsf{rect\_maxvol}(\boldsymbol{Q})$.
36     |     |   Set $\boldsymbol{X}_i = \mathsf{update\_right}(\boldsymbol{X}_{i+1}, \boldsymbol{x}_i, N_i, R_{i+1}, \boldsymbol{m}_{opt})$.
37     |   **end**
38   **end**
39   **return** $(\boldsymbol{\theta}_{min}, J_{min})$.

---

**Algorithm A.2:** Function update_left to update points of interest when traverse the tensor modes from right to left.

---

**Data:** current set of points for the $(i+1)$-th mode $\boldsymbol{X}_{i+1}$; grid points $\boldsymbol{x}_i$; number of grid points $N_i$; TT-rank $R_{i+1}$; list of indices to be selected $\boldsymbol{m}_{opt}$.

**Result:** new set of points $\boldsymbol{X}_i$.

1   Set $\boldsymbol{W}_1 = \mathsf{ones}(R_{i+1}) \otimes \boldsymbol{x}_i$ and $\boldsymbol{W}_2 = \boldsymbol{X}_{i+1} \otimes \mathsf{ones}(N_i)$
2   Set $\boldsymbol{X}_i = [\boldsymbol{W}_1, \boldsymbol{W}_2]$.
3   Select subset of rows $\boldsymbol{X}_i = \boldsymbol{X}_i[\boldsymbol{m}_{opt}, :]$.
4   **return** $\boldsymbol{X}_i$.

---

---

**Algorithm A.3:** Function update_right to update points of interest when traverse the tensor modes from left to right.

---

**Data:** current set of points for the $i$-th mode $\boldsymbol{X}_i$; grid points $\boldsymbol{x}_i$; number of grid points $N_i$; TT-rank $R_i$; list of indices to be selected $\boldsymbol{m}_{opt}$.

**Result:** new set of points $\boldsymbol{X}_{i+1}$.

1 Set $\boldsymbol{W}_1 = \mathsf{ones}(N_i) \otimes \boldsymbol{X}_i$ and $\boldsymbol{W}_2 = \boldsymbol{x}_i \otimes \mathsf{ones}(R_i)$.
2 Set $\boldsymbol{X}_{i+1} = [\boldsymbol{W}_1, \boldsymbol{W}_2]$.
3 Select subset of rows $\boldsymbol{X}_{i+1} = \boldsymbol{X}_{i+1}[\boldsymbol{m}_{opt}, :]$.
4 **return** $\boldsymbol{X}_{i+1}$.

---

---

**Algorithm A.4:** Function eval to compute the target function in points of interest.

---

**Data:** $\boldsymbol{X}_i$; $\boldsymbol{X}_{i+1}$; $R_i$; $N_i$; $R_{i+1}$; $\mathsf{J}$; $\boldsymbol{\theta}_{min}, \mathsf{J}_{min}$ (see Algorithm A.1 for details).

**Result:** transformed function values $\boldsymbol{z} \in \mathbb{R}^{R_i \cdot N_i \cdot R_{i+1}}$, updated $\boldsymbol{\theta}_{min}$ and $\mathsf{J}_{min}$.

1 Set $\boldsymbol{W}_1 = \mathsf{ones}(N_i \cdot R_{i+1}) \otimes \boldsymbol{X}_i$.
2 Set $\boldsymbol{W}_2 = \mathsf{ones}(R_{i+1}) \otimes \boldsymbol{x}_i \otimes \mathsf{ones}(R_i)$.
3 Set $\boldsymbol{W}_3 = \boldsymbol{X}_{i+1} \otimes \mathsf{ones}(R_i \cdot N_i)$.
4 Set $\boldsymbol{X}^{curr} = [\boldsymbol{W}_1, \boldsymbol{W}_2, \boldsymbol{W}_3] \in \mathbb{R}^{R_i \cdot N_i \cdot R_{i+1} \times d}$.
5 // Compute function for each point in $\boldsymbol{X}^{curr}$:
6 Set $\boldsymbol{y}^{curr} = \mathsf{J}(\boldsymbol{X}^{curr})$.
7 **if** $\min(\boldsymbol{y}^{curr}) < \mathsf{J}_{min}$ **then**
8 $\quad$ Set $m_{min} = argmin(\boldsymbol{y}^{curr})$.
9 $\quad$ Set $\boldsymbol{\theta}_{min} = \boldsymbol{X}^{curr}[m_{min}, :]$.
10 $\quad$ Set $\mathsf{J}_{min} = \boldsymbol{y}^{curr}[m_{min}]$.
11 **end**
12 // Compute smooth function for each value in $\boldsymbol{y}^{curr}$:
13 Set $\boldsymbol{z} = \frac{\pi}{2} - \arctan(\boldsymbol{y}^{curr} - \mathsf{J}_{min})$.
14 **return** $(\boldsymbol{z}, \boldsymbol{\theta}_{min}, \mathsf{J}_{min})$.

---

We then randomly initialize the TT proxy tensor with input ranks $r_{max}$. If necessary, we reduce some of the ranks to satisfy the condition $R_{i-1}N_i \geq R_i$ ($i = 1, 2, \ldots, d-1$). Note that in this case, for all TT-cores $\mathcal{G}_i \in \mathbb{R}^{R_{i-1} \times N_i \times R_i}$ ($i = 1, 2, \ldots, d$), the right unfolding matrices $\boldsymbol{G}_i^{(2)} \in \mathbb{R}^{R_{i-1} \cdot N_i \times R_i}$ will turn out to be "tall" matrices, that is, their number of rows is not less than the number of columns, and hence we can apply maxvol and rect_maxvol algorithms to these matrices.

Next, we iteratively traverse all tensor modes (using corresponding TT-cores) in the direction from right to left and vice versa. We evaluate (and transform) the objective function to refine the selected rows and columns. For each $k$-th mode of the tensor we evaluate the submatrix $\boldsymbol{J}_k^{(C)} \in \mathbb{R}^{R_{k-1} \cdot N_k \times R_k}$ of the corresponding unfolding matrix, compute its QR decomposition, find the row indices of the rectangular maximal-volume submatrix $\hat{\boldsymbol{J}}_k \in \mathbb{R}^{(R_k + \Delta R_k) \times R_k}$ of the $\mathbf{Q}$ factor and add resulting indices of the original tensor to the index set $\boldsymbol{X}_k$.

The arguments for target function evaluation in Algorithm A.4 are selected as merged left and right index sets, constructed from previous rect_maxvol computations. After each request to the objective function, we update the current optimal value $J_{min}$ and then transform the calculated values by the mapping (7) described in the main text.

## B  Additional Experiments

### B.1  Experiments with benchmark functions

In Section 3.1 we compared the TTOpt solver[3] with baseline methods, applied to various model functions [4]. The list of functions is presented in Table 1. For each function, we provide the lower/upper

---

[3]We implemented the TTOpt algorithm in a python package with detailed documentation, demos, and reproducible scripts for all benchmark calculations.

Table 1: Benchmark functions for comparison of the considered optimization algorithms and performance evaluation of the TTOpt approach. For each function, we present the lower grid bound ($a$), the upper grid bound ($b$), the global minimum ($J_{min}$) and the analytical formula. Note that $J_{min}$ for the *F6* function is given for the 10-dimensional case.

| Function | $a$ | $b$ | $J_{min}$ | Formula |
|---|---|---|---|---|
| *F1* (Ackley) | $-32.768$ | $32.768$ | $0.$ | $f(\boldsymbol{x}) = -Ae^{-B\sqrt{\frac{1}{d}\sum_{i=1}^{d}x_i^2}} - e^{\frac{1}{d}\sum_{i=1}^{d}\cos(Cx_i)} + A + e^1$, where $A = 20$, $B = 0.2$ and $C = 2\pi$ |
| *F2* (Alpine) | $-10$ | $10$ | $0.$ | $f(\boldsymbol{x}) = \sum_{i=1}^{d} \lvert x_i \sin x_i + 0.1x_i \rvert$ |
| *F3* (Brown) | $-1$ | $4$ | $0.$ | $f(\boldsymbol{x}) = \sum_{i=1}^{d-1} \left(x_i^2\right)^{(x_{i+1}^2+1)} + \left(x_{i+1}^2\right)^{(x_i^2+1)}$ |
| *F4* (Exponential) | $-1$ | $1$ | $-1.$ | $f(\boldsymbol{x}) = -e^{-\frac{1}{2}\sum_{i=1}^{d}x_i^2}$ |
| *F5* (Griewank) | $-600$ | $600$ | $0.$ | $f(\boldsymbol{x}) = \sum_{i=1}^{d} \frac{x_i^2}{4000} - \prod_{i=1}^{d} \cos\left(\frac{x_i}{\sqrt{i}}\right) + 1$ |
| *F6* (Michalewicz) | $0$ | $\pi$ | $-9.66015$ | $f(\boldsymbol{x}) = -\sum_{i=1}^{d} \sin(x_i)\sin^{2m}\left(\frac{ix_i^2}{\pi}\right)$ |
| *F7* (Qing) | $0$ | $500$ | $0.$ | $f(\boldsymbol{x}) = \sum_{i=1}^{d} \left(x_i^2 - i\right)^2$ |
| *F8* (Rastrigin) | $-5.12$ | $5.12$ | $0.$ | $f(\boldsymbol{x}) = A \cdot d + \sum_{i=1}^{d}\left(x_i^2 - A\cdot\cos(2\pi\cdot x_i)\right)$, where $A = 10$ |
| *F9* (Schaffer) | $-100$ | $100$ | $0$ | $f(\boldsymbol{x}) = \sum_{i=1}^{d-1}(0.5 + \frac{\sin^2\left(\sqrt{x_i^2+x_{i+1}^2}\right)-0.5}{\left(1+0.001(x_i^2+x_{i+1}^2)\right)^2})$ |
| *F10* (Schwefel) | $-500$ | $500$ | $0.$ | $f(\boldsymbol{x}) = 418.9829 \cdot d - \sum_{i=1}^{d} x_i \cdot \sin\left(\sqrt{\lvert x_i \rvert}\right)$ |

grid bounds ($a$ and $b$) and global minimum ($J_{min}$). Note that many benchmarks is multimodal (have two or more local optima), introducing additional complications into the optimization problem.

The main configurable parameters of our solver are the mode size ($N$; $N = 2^q$ in the case of quantization, where $q$ is the number of submodes in the quantized tensor); the rank ($R$), and the limit on the number of requests to the objective function ($M$). The choice of these parameters can affect the final accuracy of the optimization process. Below we present the results of the studies of parameter importance. In all calculations, we fixed the non-varying parameters at the values provided in Section 3.1.

**Mode size influence.** To reach high accuracy, we need fine grids. As we indicated in Section 2.6, in this case, the quantization of the tensor modes seems attractive. We reshape the original $d$-dimensional tensor $\mathcal{J} \in \mathbb{R}^{N_1 \times N_2 \times \cdots \times N_d}$ into the tensor $\tilde{\mathcal{J}} \in \mathbb{R}^{2 \times 2 \times \cdots \times 2}$ of a higher dimension $d \cdot q$, but with smaller modes of size 2, and apply the TTOpt algorithm to this "long" tensor instead of the original one.

Table 2: Comparison of the "direct" (TT) and "quantized" (QTT) TTOpt solvers in terms of the final error (absolute deviation of the obtained value from the exact minimum) for various benchmark functions. The reported values are averaged over ten independent runs.

| MODE SIZE | | F1 | F2 | F3 | F4 | F5 | F6 | F7 | F8 | F9 | F10 |
|---|---|---|---|---|---|---|---|---|---|---|---|
| 256 | TT | 1.2E+00 | 2.0E-02 | 0.0E+00 | 7.7E-05 | 1.0E+00 | 9.8E-02 | 9.4E+01 | 8.0E-01 | 4.2E-01 | 4.5E-01 |
| | QTT | 1.2E+00 | 2.3E-02 | 0.0E+00 | 7.7E-05 | 1.0E+00 | 1.6E-01 | 9.4E+01 | 8.0E-01 | 3.9E-01 | 4.5E-01 |
| 1024 | TT | 1.6E+01 | 4.2E+00 | 3.0E+01 | 2.6E-01 | 5.7E+01 | 2.0E+00 | 1.9E+10 | 4.7E+01 | 1.5E+00 | 1.1E+03 |
| | QTT | 1.8E-01 | 8.2E-03 | 6.9E-05 | 4.8E-06 | 2.1E-01 | 7.1E-02 | 5.2E+00 | 5.0E-02 | 1.2E-01 | 2.8E-02 |
| 4096 | TT | 1.9E+01 | 1.5E+01 | 5.0E+08 | 5.1E-01 | 1.4E+02 | 5.7E+00 | 5.1E+10 | 9.8E+01 | 3.5E+00 | 2.6E+03 |
| | QTT | 3.5E-02 | 1.8E-03 | 0.0E+00 | 3.0E-07 | 3.9E-02 | 4.3E-02 | 2.6E-01 | 3.1E-03 | 8.7E-02 | 1.0E-02 |
| 16384 | TT | 2.0E+01 | 1.9E+01 | 9.9E+17 | 5.9E-01 | 1.7E+02 | 5.9E+00 | 6.3E+10 | 1.2E+02 | 3.8E+00 | 3.2E+03 |
| | QTT | 8.2E-03 | 7.8E-04 | 2.7E-07 | 1.9E-08 | 2.6E-02 | 8.9E-02 | 2.2E-02 | 1.9E-04 | 1.2E-01 | 3.8E-04 |
| 65536 | TT | 2.0E+01 | 1.9E+01 | 1.2E+10 | 6.7E-01 | 2.2E+02 | 7.8E+00 | 7.1E+10 | 1.6E+02 | 4.4E+00 | 3.6E+03 |
| | QTT | 2.0E-03 | 1.3E-04 | 1.2E-09 | 1.2E-09 | 2.2E-02 | 7.1E-02 | 9.4E-04 | 1.2E-05 | 1.4E-01 | 1.6E-04 |
| 262144 | TT | 2.1E+01 | 2.3E+01 | 1.7E+16 | 8.0E-01 | 3.0E+02 | 8.4E+00 | 8.9E+10 | 1.8E+02 | 4.5E+00 | 3.7E+03 |
| | QTT | 5.0E-04 | 3.3E-05 | 1.1E-09 | 7.3E-11 | 3.0E-02 | 3.8E-02 | 3.7E-05 | 7.6E-07 | 1.4E-01 | 1.3E-04 |
| 1048576 | TT | 2.1E+01 | 2.8E+01 | 5.5E+16 | 8.3E-01 | 3.3E+02 | 8.4E+00 | 1.2E+11 | 1.9E+02 | 4.4E+00 | 3.7E+03 |
| | QTT | 1.3E-04 | 1.2E-05 | 0.0E+00 | 4.5E-12 | 1.7E-02 | 6.8E-02 | 5.9E-06 | 4.7E-08 | 1.1E-01 | 1.3E-04 |

In Table 2 we present the comparison of optimization results for the basic algorithm without quantization ("TT") and for the improved algorithm with quantization ("QTT"). For each value $N$ of the mode size, we choose the number of submodes in the quantized tensor as $q = \log_2 N$. The QTT-solver gives several orders of magnitude more accurate results than the TT-solver. At the same time, for the QTT-solver, a regular decrease in the error is observed with an increase in the mode size. Thus, for the stable operation of gradient-free optimization methods based on the low-rank tensor approximations, it is necessary to quantize the modes of the original tensor.

**Rank influence.** The rank (the size of the maximal-volume submatrices) determines how many points are queried at each iteration of the TTOpt algorithm, and this parameter is similar to population size in evolutionary algorithms. Small maximal-volume submatrices may give a better bound for maximal elements (see Eq. (3) from the main text), but finding small submatrices may be more challenging for the algorithm and may lead to numerical instabilities. At the same time, when choosing rank $R$, we should take into account that the algorithm will need $2 \cdot T \cdot (dq) \cdot P \cdot R^2$ function calls, where $T$ is the number of sweeps (it should be at least 1, however, for better convergence, it is worth taking values of $4 - 5$) and $P = 2$ is a submode size. Hence we have inequality $R \leq \sqrt{\frac{M}{4 \cdot T \cdot d \cdot q}}$, where $M$ is a given limit on the number of function requests.

In Figure 1 we demonstrate the dependence of the TTOpt's accuracy on the rank. As can be seen, with small ranks (1 or 2), we have too low accuracy for most benchmarks. At the same time, the accuracy begins to drop at too high-rank values (7 or more), which is due to the insufficient number of sweeps taken by the algorithm for convergence.

**Number of function queries influence.** The number of requests to the objective function can be determined automatically based on algorithm iterations. Thus, with a total number of sweeps $T$, we will have $\mathcal{O}\left(T \cdot d \cdot \max_{1 \leq k \leq d}\left(N_k R_k^2\right)\right)$ calls to the objective function. However, in practice, it turns out to be more convenient to limit the maximum number of function calls, $M$, according to the computational budget.

In Figure 2 the dependence of the accuracy on the total number of requests, $M$, to the objective function is presented. Predictably, as $M$ increases, the accuracy also increases. The plateau for benchmarks are associated with the dependence of the result on the remaining parameters $(R, q)$ of the TTOpt solver.

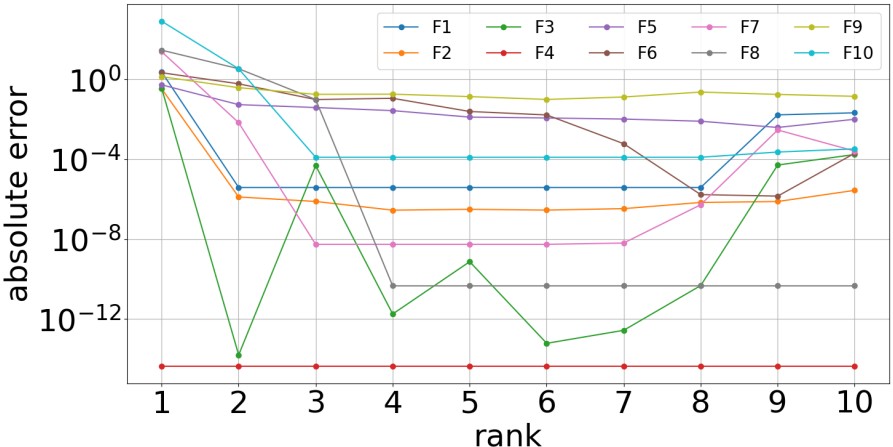

Figure 1: The dependence of the final error (absolute deviation of the obtained value from the exact minimum) on the rank for various benchmark functions. The reported values are averaged over ten independent runs.

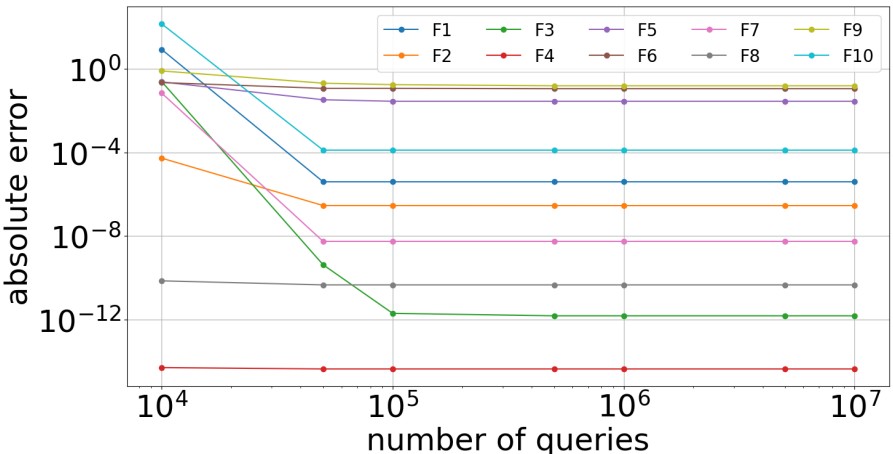

Figure 2: The dependence of the final error (absolute deviation of the obtained value from the exact minimum) on the number of target functions calls for various benchmark functions. The reported values are averaged over ten independent runs.

**Function dimensionality influence.** One of the advantages of the proposed approach is the possibility of its application to essentially multidimensional functions. In Table 3 we present the results of TTOpt for functions of various dimensions (we removed the *F6* function from benchmarks, since its optima are known only for 2, 5 and 10-dimensional cases). Note that as a limit on the number of requests of the objective function, we choose $10^4 \cdot d$, and the values of the remaining parameters were chosen the same as above.

As can be seen, even for 500-dimensional functions, the TTOpt method results in fairly accurate solutions for most benchmarks. However, for benchmarks *F4*, *F7* and *F9* the errors are larger than for lower dimensions. We suspect that our heuristic of the number of objective function evaluations is not accurate in these cases.

Table 3: The result of the TTOpt optimizer in terms of the final error $\epsilon$ (absolute deviation of the obtained optimal value relative to the global minimum) and computation time $\tau$ (in seconds) for various benchmark functions and various dimension numbers ($d$).

| FUNCTION | | $d = 10$ | $d = 50$ | $d = 100$ | $d = 500$ |
|---|---|---|---|---|---|
| F1 | $\epsilon$ | 3.9E-06 | 3.9E-06 | 3.9E-06 | 3.9E-06 |
| | $\tau$ | 3.1 | 37.8 | 131.0 | 3153.5 |
| F2 | $\epsilon$ | 2.9E-07 | 3.7E-06 | 5.2E-06 | 2.1E-05 |
| | $\tau$ | 2.5 | 36.3 | 129.5 | 3153.1 |
| F3 | $\epsilon$ | 2.3E-12 | 4.9E-10 | 1.1E-09 | 4.7E-09 |
| | $\tau$ | 2.6 | 36.8 | 132.6 | 3205.4 |
| F4 | $\epsilon$ | 4.4E-15 | 2.2E-14 | 4.4E-14 | 1.0E+00 |
| | $\tau$ | 2.5 | 35.6 | 129.2 | 3131.1 |
| F5 | $\epsilon$ | 2.5E-02 | 3.7E-02 | 3.7E-02 | 3.7E-02 |
| | $\tau$ | 2.5 | 36.1 | 130.4 | 3132.2 |
| F7 | $\epsilon$ | 5.5E-09 | 8.9E-08 | 3.4E-07 | 5.6E+02 |
| | $\tau$ | 2.5 | 35.6 | 130.2 | 3123.7 |
| F8 | $\epsilon$ | 4.6E-11 | 2.3E-10 | 4.6E-10 | 2.3E-09 |
| | $\tau$ | 2.5 | 35.6 | 130.1 | 3124.0 |
| F9 | $\epsilon$ | 3.4E-01 | 9.3E-01 | 2.2E+00 | 1.0E+01 |
| | $\tau$ | 2.5 | 36.0 | 130.6 | 3157.4 |
| F10 | $\epsilon$ | 1.3E-04 | 6.4E-04 | 1.3E-03 | 6.4E-03 |
| | $\tau$ | 2.6 | 35.7 | 130.0 | 3140.7 |

Table 4: Comparison of the TTOpt optimizer with Bayesian optimization [6] baselines in terms of the final error $\epsilon$ (absolute deviation of the obtained optimal value relative to the global minimum) and computation time $\tau$ (in seconds) for various 10-dimensional benchmark functions. Note that $\tau$ values for Simultaneous Optimistic Optimization (SOO), Direct Simultaneous Optimistic Optimization (dSOO), Locally Oriented Global Optimization (LOGO) and Random Optimization (RANDOM) refers to the time measured for a complied C-code, while our TTOpt optimizer is implemented in python, and will be more time-efficient if written in C.

| | | ACKLEY | RASTRIGIN | ROSENBROCK | SCHWEFEL |
|---|---|---|---|---|---|
| TTOPT | $\epsilon$ | 3.9E-06 | **4.6E-11** | **3.9E-01** | **8.4E-02** |
| | $\tau$ | 1.23 | 1.21 | 1.18 | 1.21 |
| dSOO | $\epsilon$ | **4.0E-10** | 2.0E+00 | 8.1E+00 | 5.3E+02 |
| | $\tau$ | 8.10 | 7.20 | 7.75 | 7.01 |
| SOO | $\epsilon$ | 9.0E-10 | 2.29E+00 | 7.0E+02 | 5.2E+02 |
| | $\tau$ | 7.44 | 7.57 | 0.66 | 7.31 |
| LOGO | $\epsilon$ | 1.2E-09 | 3.44E+01 | 7.9E+00 | 5.3E+02 |
| | $\tau$ | 6.80 | 0.77 | 7.06 | 7.51 |
| RANDOM | $\epsilon$ | 1.1E+01 | 2.29E+00 | 1.5E+00 | 1.2E+03 |
| | $\tau$ | 0.78 | 7.34 | 7.25 | 0.77 |

## B.2 Comparison with Bayesian optimization

In Table 4 we present the results of TTOpt and several Bayesian methods for 10-dimensional benchmarks. We selected functions supported by the Bayesian optimization package from[4] [6]. Note that in all cases we chose $10^5$ as the limit on the number of requests to the objective function and the values of the remaining parameters were chosen the same as above. TTOpt outperforms all tested Bayesian algorithms for Rastrigin, Rosenbrock, and Schwefel functions. For the Ackley function, the difference in accuracy is not significant. On average, TTOpt is faster than Bayesian methods, despite they are implemented in **C** language. We stress that standard Bayesian methods are not applicable in higher-dimensional problems.

## B.3 Formulation of reinforcement learning problem as black-box optimization task

Here we describe a typical reinforcement learning setting within Markov decision process formalism. The agent acts in the environment that has a set of states $S$. In each state $s \in S$ the agent takes an action from a set of actions $a \in A$. Upon taking this action, the agent receives a local reward $r(s, a)$ and reaches a new state $s'$, determined by the transition probability distribution $\mathcal{T}(s' \mid s, a)$. The policy $\pi(a \mid s)$ specifies which action the agent will take depending on its current state. Upon taking $T$ ($T$ is also called horizon) actions, the agent receives a cumulative reward, defined as

$$J = \sum_{t=0}^{T-1} \gamma^t r(s_t, a_t), \tag{3}$$

where $\gamma \in [0, 1]$ the is discounting factor, specifies the relevance of historic rewards for the current step.

The goal of the agent is to find the policy $\pi^*(a \mid s)$ that maximizes the expected cumulative reward $J$ over the agent's lifetime. In policy-based approaches, the policy is approximated by a function $\pi(a \mid s, \boldsymbol{\theta})$ (for example, a neural network), which depends on a vector of parameters $\boldsymbol{\theta}$. It follows then that the cumulative reward is a function of the parameters of the agent:

$$J(\boldsymbol{\theta}) = \mathbb{E}_{(s_t, a_t) \sim \mathcal{T}, \pi(\boldsymbol{\theta})} \left[ \sum_{t=0}^{T-1} \gamma^t r(s_t, a_t)) \right], \tag{4}$$

where $r(s_t, a_t) \sim r(s_t, \pi(s_{t-1} \mid \boldsymbol{\theta}))$. In case of episodic tasks we can assume $\gamma = 1$. Finding an optimal policy can be done by maximizing the cumulative reward $J$ with respect to parameters $\boldsymbol{\theta}$:

$$\pi^*(a \mid s) = \pi(a \mid s, \boldsymbol{\theta}^*), \tag{5}$$

where $\boldsymbol{\theta}^* \simeq \text{argmax } J(\boldsymbol{\theta})$. Notice that $J$ may be non-differentiable due to the stochastic nature of $\mathcal{T}$ or the definition of $r$, depending on a particular problem formulation. However, this does not pose a problem for direct optimization algorithms.

To summarize, the RL problem can be transformed into a simple optimization problem for the cumulative reward $J(\boldsymbol{\theta})$. The parameters of this function are the weights of the agent. Optimization of the cumulative reward with direct optimization algorithms is an on-policy learning in RL algorithm classification.

## B.4 Rank dependence study

Since rank is an important parameter of our method, we studied its influence on the rewards in RL, see Figure 5. Note that the rank determines how many points are queried at each iteration, and this parameter is similar to population size in evolutionary algorithms. We found almost no dependency of the final reward on rank after $R > 3$ (on average). The Eq. (3) from the main text states that small maximal-volume submatrices should give a better bound for the maximal element. However, finding small submatrices may be more challenging for the algorithm. It turns out that reward functions in considered RL tasks are "good" for the maximum volume heuristic, e.g., even with small ranks, the algorithm produces high-quality solutions.

---

[4]The source code is available at `https://github.com/Eiii/opt_cmp`

Table 5: The mean and standard deviation ($\mathbb{E} \pm \sigma$) of final cumulative reward before and after fine-tuning with TTOpt. The policy's weights are from the original repository of ARS [5].

|  | ARS [5] | ARS TTOPT($2^8$) |
|---|---|---|
| ANT-V3 | 4972.48±21.58 | **5039.90**±57.00 |
| HALFCHEETAH-V3 | 6527.89±82.70 | **6840.39**±87.41 |
| HOPPER-V3 | **3764.74**±355.08 | 3296.49±11.81 |
| HUMANOID-V3 | 11439.79±51.44 | **11560.01**±54.08 |
| SWIMMER-V3 | 354.43±2.32 | **361.87**±1.76 |
| WALKER2D-V3 | **11519.77**±112.55 | 11216.25±88.32 |

Table 6: The number of hidden units in each layer of convolutional policy $h$, the total number of parameters $d$, the sizes of the state and action spaces $A$ and $S$, the rank $R$ and the activation function between the layers (Act.). The average number of function quires per iteration (population size) in the case of TTOpt and ES baselines, respectively, is denoted by $Q$ (the values separated by a comma). The number of seeds is $S_d$.

|  | $H$ | $D$ | $S$ | $A$ | $R$ | $Act.$ | $Q$ | $S_d$ |
|---|---|---|---|---|---|---|---|---|
| S | 8 | 55 | 8 | 2 | 3 | TANH | 55, 64 | 7 |
| L | 8 | 55 | 8 | 2 | 3 | RELU | 53, 64 | 7 |
| I | 4 | 26 | 4 | 1 | 3 | TANH | 57, 64 | 7 |
| H | 4 | 44 | 17 | 6 | 5 | TANH | 120, 128 | 7 |

## B.5 Constraint Handling in Evolutionary Algorithms

There are two options to satisfy constraints in evolutionary computation called projection and penalization. These steps can be represented as two functions, $\boldsymbol{\theta}_p = f_{proj}(\boldsymbol{\theta})$, and $f_{pen}(\boldsymbol{\theta}_p, \boldsymbol{\theta})$ with a regularization term:

$$J_p(\boldsymbol{\theta}) = J(\boldsymbol{\theta}_p) - \lambda f_{pen}(\boldsymbol{\theta}_p, \boldsymbol{\theta}). \tag{6}$$

In this work, we use the constraint functions described below. *CDF projection* is applied in experiments with mode size $N = 3$ (see Table 3 from the main text). The idea is to use the cumulative density function to map normally distributed parameters of the policies to {-1,0,1} set:

$$\boldsymbol{\theta} = \begin{cases} -1 & \text{CDF}(\boldsymbol{\theta}) \leq \frac{1}{3}, \\ 0 & \frac{1}{3} < \text{CDF}(\boldsymbol{\theta}) < \frac{2}{3}, \\ 1 & \text{CDF}(\boldsymbol{\theta}) \geq \frac{2}{3}. \end{cases} \tag{7}$$

*Uniform projection* is applied when $N = 256$ in experiments shown in Table (3) from the main text. In this case, the idea is to keep the value if it satisfies the bounds, otherwise, we draw a new sample uniformly from a grid defined in Algorithm A.1:

$$\theta_p^i = \begin{cases} \theta^i, & \text{if } L \leq \theta^i \leq U, \\ \mathbf{x}_i[k] & \text{otherwise.} \end{cases} \tag{8}$$

*Quadratic penalty* is applied in all experiments. If $L$ and $U$ are the bounds, then $f_{pen}(\boldsymbol{\theta}_p, \boldsymbol{\theta}) = \sum_{i:\theta^i<L}(L - \theta^i)^2 + \sum_{i:\theta^i>U}(\theta^i - U)^2$. We set $\lambda = 0.1$ in all experiments.

## B.6 Reinforcement Learning Experiments

Figure 3 and Figure 4 show training curves for all test environments which were not included in the main text.

**Fine-tuning of Linear Policies.** We use TTOpt to fine-tune Augmented Random Search (ARS) [5] linear policies obtained from the original paper. The cost function is the average of seven independent episodes with fixed random seeds. The upper and lower grid bounds are estimated using statistics of pre-trained linear policies: $b_i = \theta_i \pm \alpha \cdot \sigma(\boldsymbol{\theta})$ with $\alpha = 0.1$. For Ant, Humanoid [2], Walker [2] and HalfCheetah [9] we select $\alpha = 0.5$, and for Swimmer [1] and Hopper [8] we set $\alpha = 1$.

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

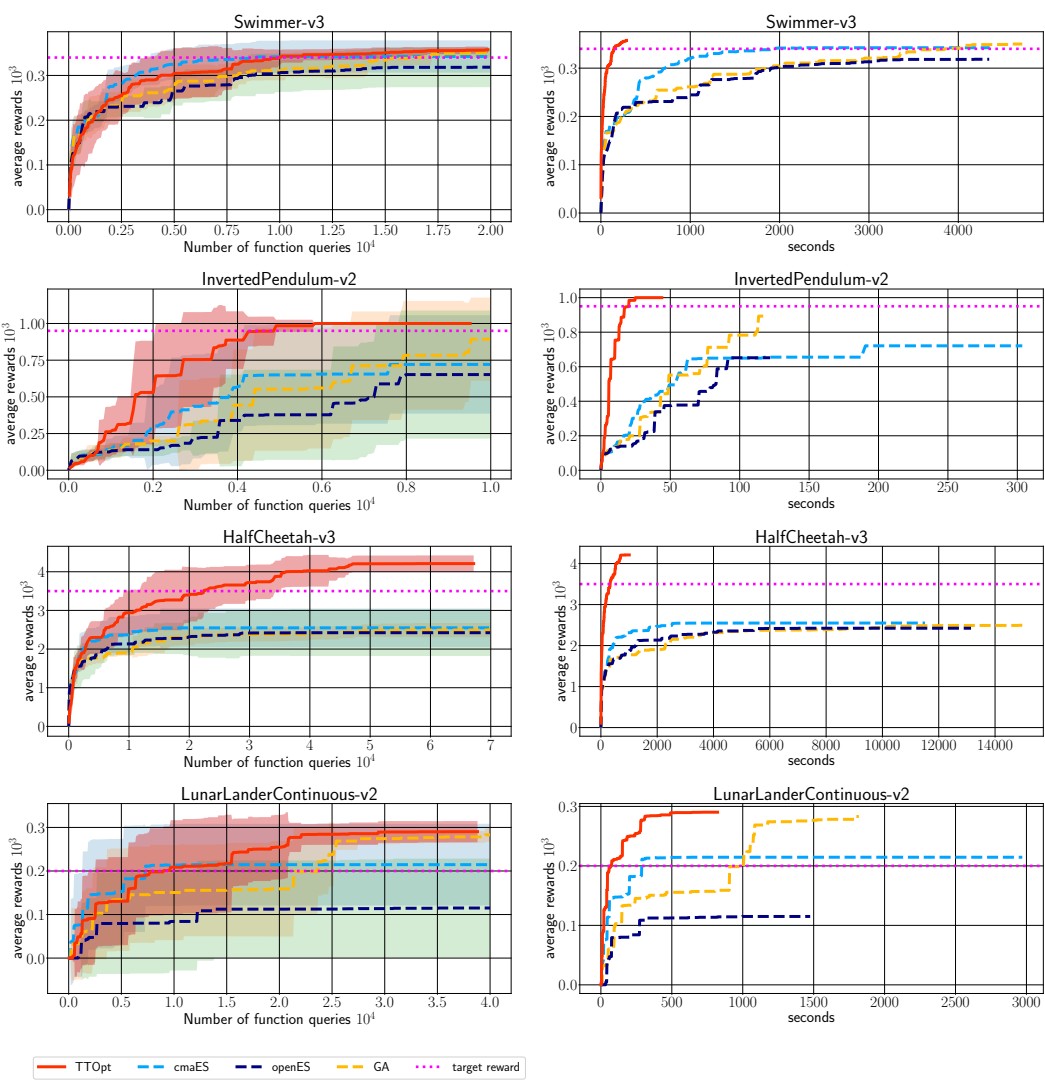

Figure 3: Training curves of TTOpt and baselines for $N = 3$ possible weight values: $(-1, 0, 1)$. Left is the dependence of the average cumulative reward on the number of interactions with the environment (episodes). Right is the same reward depending on the execution time. The reward is averaged for seven seeds. The shaded area shows the difference of one standard deviation around the mean.

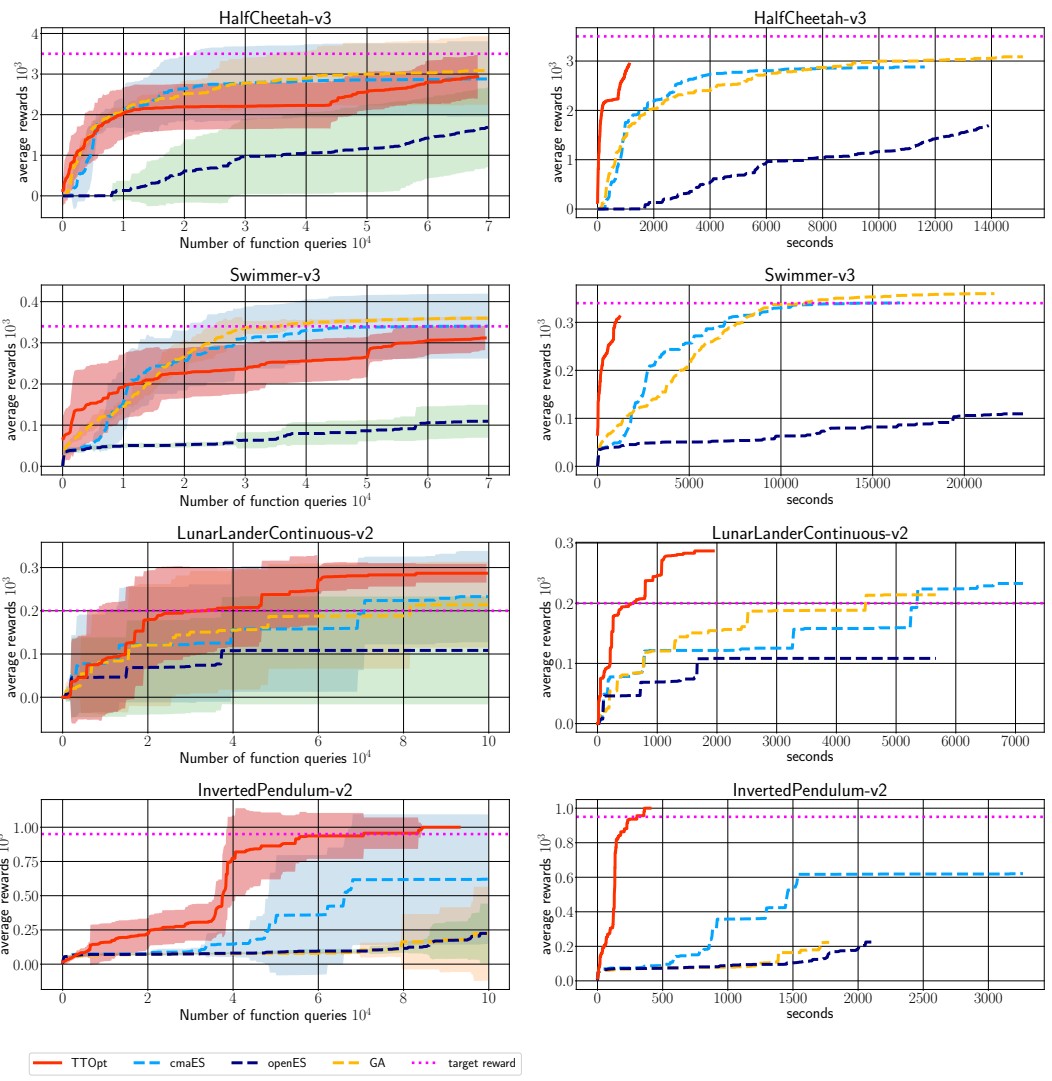

Figure 4: Training curves of TTOpt and baselines for $N = 256$ possible weight values. Left is the dependence of the average cumulative reward on the number of interactions with the environment (episodes). Right is the same reward depending on the execution time. The reward is averaged for seven seeds. The shaded area shows the difference of one standard deviation around the mean.

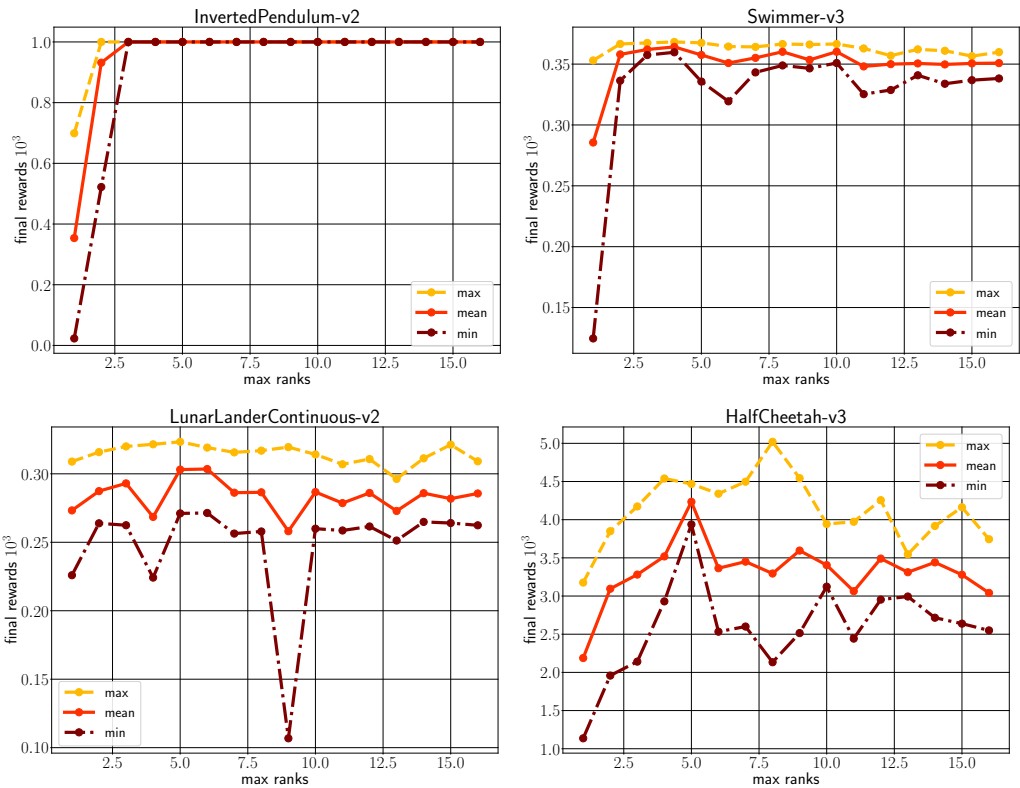

Figure 5: The dependency of the final cumulative reward on rank. The mean, the minimum, and the maximum over seven random seeds are presented. The mode size is $N = 3$.