# OpenReview forum: "TTOpt: A Maximum Volume Quantized Tensor Train-based Optimization and its Application to Reinforcement Learning"
_NeurIPS.cc/2022/Conference — NeurIPS 2022 Accept_

### Official Review · Reviewer_WXFn · 2022-07-05

**Rating:** 6
**Confidence:** 3
**Soundness:** 3 good
**Presentation:** 3 good
**Contribution:** 2 fair

**Summary:**

The paper proposes an optimization method for multivariate functions of discrete arguments by considering their low-rank tensor decomposition in terms of Tensor Train structure. The method is compared to gradient-free methods, and is evaluated on Reinforcement Learning tasks and on functions with continuous arguments.

**Questions:**

* Questions
	- Maximum Value in matrix is bounded by $\hat{J}_{max} \geq \frac{J}{R^2}$, for tensors with $d$ indices, would this bound grow to the power of $d$, doesn't this bound increase with the compression trick?
	- What happens when you have a tensor train of $d=2$, where there are two minimal values of $-1$ in each, and a maximal value of $0.5$ in each? the total maximal value is $1$, isn't it? (The mapping function in equation 7 is applied at the end)
	- Why are the values in Table 2 in the appendix are of order of magnitude different than presented in Table 1 of the main text? of instance, the absolute deviation from the minimum of the Ackley function is $1.2$ (the absolute minimum is at $0$), but the relative deviation is $3.9e-6$.
    - It seems all the test functions have the symmetry of $x_i \rightarrow -x_i$. How would the method work for non-symmetric test functions?


* Suggestions
	- Perhaps add the definition for "Maximum modulus element" to the main text.
	- Perhaps add an evaluation of the method with the compression to modes and without (line 178).
	- It would be interesting to see how the approach handles non-differentiable functions as it is gradient-free.


* Minor corrections:
  - line 181 "Let assume" -> Assume that
  - Appendix Table 2 Grienwank -> Griewank

**Limitations:**

* I believe that when the method is applied to continuous functions, it is sensitive to the chosen discretization grid. Moreover, the complexity of the method increases for finer grids, as the $N_k$ increases.

**Strengths And Weaknesses:**

The main text is clear and well written. This work combines various solutions to solve this optimization problem, that is, extending the maxvol function to higher order tensors, replacing high dimensions by decomposing them to lower ones, and applying an updated mapping function to deal with algorithmic constraints. Additional experiments  testing different hyper-parameters appear in the appendix. The paper lacks, however, experiments that demonstrate the contribution of each individual part of the method empirically, such as the mapping function and the mode compression.

If the landscape of RL consisted of only of gradient-free methods, then the presented low execution time of the method, together with its ability to solve the problems successfully, makes this work significant. The fact there is no comparison to gradient-based methods is a major weakness and complicates the assessment of the possible impact of this work.

Strengths:
* Extends a matrix-based method to higher order tensors.
* Low execution time in comparison to other gradient-free approaches.
* Various experiments regarding the hyperparameters of the method appear in the appendix.

Weaknesses:
* No comparison with gradient-based methods.
* The comparison to RL is lacking. There is no comparison to any gradient-based method.
* There are no ablations to the benefits of using either the mapping functions (Eq. 7) or mode compression.

---

> ### Author Response · Authors · 2022-08-01
> **Weakness. No comparison with gradient-based methods ...**
>
> We agree that gradient-based methods are powerful, but as baselines, we use black-box optimization methods since they utilize the same information as TTOpt. Comparison to gradient-type methods has been done in prior works (for instance, see https://arxiv.org/pdf/1803.07055.pdf). An additional property of gradient-free methods is the ability to handle integer/binary variables and quantized weights. For example, in RL experiments we learn policy with binary weights, which can not be done by policy gradient-type methods directly.
>
> To address the reviewer's concerns, we performed preliminary experiments with Proximal Policy Optimization (PPO) method. Note that this comparison is not completely fair: the PPO method used floating point weights, while gradient-free methods use weights with values $\{-1, 0, 1\}$. We limit the number of episodes to $10^5$. The results are shown in the table below. In this setup, PPO is less efficient than TTOpt in Swimmer, Inverted Pendulum, and Half-Cheetah environments, while PPO wins in the Lunar Lander environment. The run time of PPO is larger than TTOpt in all cases. Please note that these results are provided for quick checks only.
>
> ---
>
> An additional comparison of the TTOpt method and baselines presented in the paper (see Tab. 3) with the Proximal Policy Optimization method (PPO; Schulman et al. 2017) under the same budget up to $10^5$ episodes. The top table shows the mean and standard deviation of the final cumulative reward. The bottom table presents the calculation time in seconds:
>
> | Method   | S             | L              | I              | H                |
> |:-------------|:--------------|:---------------|:---------------|:-----------------|
> | PPO          | 95.63±40.59  | 303.83±2.75   | 815.38±369.23 | 3196.88±1187.82 |
> | TTopt (ours)  | 357.50±6.59  | 290.29±24.40  | 1000.00±0.00  | 4211.02±211.94  |
> | cmaES       | 342.31±36.07 | 214.55±93.79  | 721.00±335.37 | 2549.83±501.08  |
> | simpleGA    | 349.91±10.04 | 283.05±16.28  | 893.00±283.10 | 2495.37±185.11  |
> | openES      | 318.39±44.61 | 114.97±113.48 | 651.86±436.37 | 2423.16±602.43  |
>
> |    |      PPO |     TTOpt |
> |:---|---------:|----------:|
> | S  | 28670  |  287  |
> | L  | 19398  |  823  |
> | I  |  2238 |   44 |
> | H  | 96050  | 1008   |

---

> ### Author Response · Authors · 2022-08-01
> **Weakness. There are no ablations to the benefits of using either the mapping functions or mode compression.**
>
> Thank you for raising this point. Please note that we provided an ablation study for the mode compression in Tab. 2 in Supplementary for benchmark functions. The use of mode compression (i.e., QTT) leads to many orders of magnitude improvements in the accuracy of the method. As can be inferred from this table, the larger the grid size, the more drastic the effect of mode compression for most functions.
>
> The use of the mapping function is an integral part of the TTOpt algorithm and it can not be omitted, hence no ablation study is possible. This function has to transform the minimal element to the maximal by modulus element and should be continuous, smooth, and strictly monotone. We experimented with $g(x) = \frac{\pi}{2} - atan(J(x)-J_{min})$ and $g(x) = e^{\alpha \cdot (J(x) - J_{min})}$, but found no influence of this choice on the performance of the method.

---

> ### Author Response · Authors · 2022-08-01
> **Question 1. ... bound grow to the power of d ...**
>
> Thank you for raising this point. With the transition from TT to QTT format, the value of the rank may increase, but it is not possible to make exact estimates. Our numerical experiments show that the use of QTT (with larger $d$) leads to a significant increase in the accuracy (see Tab. 2 in Supplementary). In most cases, the accuracy of QTT grows significantly with the grid size.

---

> ### Author Response · Authors · 2022-08-01
> **Question 2. ... what happens when you have a tensor train of d=2 ...**
>
> If we correctly understand the question, you are asking to explain why we need a mapping function. As an example, consider a matrix with the minimal element $-1$ and a maximal element $0.5$. The TTOpt algorithm will find the maximal by modulus element $-1$ (the minimum) even without the mapping function. However, suppose that the minimum is $-0.25$. Then the maximal by modulus element in this matrix is $0.5$, which is not the minimum. The mapping function allows the algorithm to find the minimum by transforming
>     it to largest value. Assume that the current estimate of the minimum is 0, then
>     $g(0.5, 0) = \frac{\pi}{2} - atan(0.5-0) < g(-0.25, 0) = \frac{\pi}{2} - atan(-0.25-0)$. After this mapping, TTOpt can correctly identify $-0.25$ as the minimum.

---

> ### Author Response · Authors · 2022-08-01
> **Question 3. ... Why are the values in Table 2 in the appendix of an order of magnitude different ...**
>
> Thank you for your question, we will try to clarify this point. Please note that Tab. 1 in the main text and Tab. 2 in the Supplementary list results in a completely different choice of grid size. The result in the main text uses a grid of size $2^{25}$ in each mode for the QTTOpt method. In Tab. 2 in Supplementary, we compare results for grids of sizes from $2^8$ to $2^{20}$. Tab. 2 demonstrates the drastic effects of using QTT (mode compression) compared to the naive application of TT. The last result for Ackley function in Tab. 2 in Supplementary ($1.3E-04$, $2^{20}$ grid size) is sufficiently close to the value in Tab. 1 in the main text ($3.9E-06$, $2^{25}$ grid size). Note that all values are absolute errors.

---

> ### Author Response · Authors · 2022-08-01
> **Question 4. ... how would the method work for non-symmetric test functions?**
>
> Thanks to this observation, indeed many of the benchmark functions have the mentioned symmetry. Note, however, that Schwefel function (_F10_) is not symmetric, and the Alpine function (_F2_) is not differentiable. In both of these cases we obtained the lowest errors compared to all baselines (see Tab. 1). For convenience, we added analytic expressions of benchmark functions to Tab. 1 in the Supplementary.

---

> ### Author Response · Authors · 2022-08-01
> **Suggestions and Minor corrections.**
>
> Thank you, we implemented the proposed corrections. Some of these points were addressed in previous responses.

---

> ### Author Response · Authors · 2022-08-01
> **Limitations. ... it is sensitive to the chosen discretization grid ...**
>
> Indeed, you are correct about the dependence on the discretization grid. The advantage of TTOpt is that very fine grids can be used. For most experiments with benchmark functions, we used $2^{25}$ grids. The study of the effect of grid size can be found in Tab. 2 in Supplementary.

---

> ### Comment · Reviewer_WXFn · 2022-08-07
> **Post Rebuttal**
>
> Thank you for your clarifications and willingness to run additional experiments.
>
> * Your answer on the error bound when mode compression is applied is still not clear to me.
> * I still think gradient-descent baseline for the function minima estimation is important. (Table 1)
> * The Schwefel function is symmetric with respect to permutations of the different entries of $x_i$, nevertheless, I find it one of the most challenging functions that were tested, as its global minimum is not located at the origin. I believe the high absolute error (in comparison to the other functions) is due to discretization.
> * The alpine function is indeed no differentiable, however, it has multiple minima with the same value. I wonder how it would behave if the origin is not included in its discretization grid.

---

> > ### Author Response · Authors · 2022-08-08
> > **General comments**
> >
> > Thank you for your comments! We conducted additional numerical experiments using gradient-based methods. We add the results to the main text (see Tab. 1 with 7 new methods/rows and explanations on LL. 198-202, 215-216, 223-226). As it turned out, for 6 benchmarks, TTOpt gives a significantly more accurate result for the same number of requests to the objective function. Compared to other methods, TTOpt is consistently accurate and avoids random failures to converge.
> >
> > In our comment on the error bound, we meant that the compression trick does not formally change the corresponding estimate (eq. 3), but the rank value (R) in eq. 3 may turn out to be different. However, it should be noted that at the moment there is no formal proof substantiating the applicability of eq. 3 in the multidimensional case (since we never build any unfolding matrix in full during TTOpt iterations), but this formula is a motivation for the maxvol-like updates, which we perform during the sweeps. The only guarantee is that the result will monotonically improve with iterations. In the same time, our algorithm works very well in practice for a large class of difficult and widely tested high dimensional benchmarks for global optimization and in most cases close to optimum solutions are found. Thus, our experiments show that this mathematical bound is not very tight and rather pessimistic.

---

### Official Review · Reviewer_uApo · 2022-07-09

**Rating:** 7
**Confidence:** 4
**Soundness:** 3 good
**Presentation:** 2 fair
**Contribution:** 4 excellent

**Summary:**

The paper presents a gradient-free optimization method, based on the tensor train approximation of the functional and the maximum volume principle, adapted to navigate the optimized parameter space instead of reconstructing the volume. Experiments on multidimensional analytical functions and reinforcement learning settings demonstrate superior convergence rates and efficiency in terms of calls to the black-box function.

**Questions:**

1. L47 "Our algorithm can directly train quantized neural networks" - can the authors explain this? I understand the concept of QTT, but this point is only seen in the introduction and slightly touched around the RL part. Also, L235 "These experiments model the case of quantized neural networks which use q-bits quantization." - please elaborate on how NN parameter quantization relates to TT mode quantization.
2. In relation to Table 3, please discuss why some environments benefit from finer quantization while others don't, especially given the low variance of curves?
3. What are the convergence guarantees of TTopt?
4. The paper mentions "up to machine precision" (L40), and by the look of fp exponent of results in Table 1 and on, single-precision was used. Do the results improve further with double precision?
5. In Algorithm A1, rank selection only accounts for the left-hand side constraint on the ranks but does not enforce the right-hand side constraint. Shouldn't it also be the $\min(R_{i-1} N_i, R_i N_i, r_{max})$? This obviously cannot be done in a single loop due to double-side recursion; however, two loops can do. Wouldn't it be better than the reduction procedure from LL58-61 of the supplementary?
6. In Algorithm A1, "Set $G_i = \mathrm{random}(R_i N_i , R_{i+1})$, random how? Does the distribution matter?
7. Why is the case of linear policies important to consider TTopt with?

Writing tips:
- Order of environments in the left and right parts of Table 3 is different for no good reason
- Consider enabling line numbering built into algorithmicx for ease of referring to line numbers
- Annotating algorithms with shapes of operands would help in understanding
- Consider prioritizing a self-contained intro into the experimental domains and smooth explanation of TTopt plugging into the big picture, over explanations of TT and maxvol, provided that the main paper cannot fit the core algorithms anyway due to limited space.

**Limitations:**

Answers to several questions asked above, in combination with the rather deep analysis of rank and number of evaluation dependencies,  cover the most interesting limitation scenarios.

**Strengths And Weaknesses:**

Judging by the provided analysis and results, TTopt deserves membership in most gradient-free optimization toolboxes.
Extending maxvol principles for optimization and mandatory quantization of the continuous case are interesting, novel, and significant (conditioned on the presented results).
The study of TTopt on analytical functions is rich and deep, going as far as comparing with Bayesian optimization methods.

However, given the impressive results from Fig.3 in RL environments, too little space is allocated to explaining the setup and discussion of what makes two orders of magnitude improvement in reward possible.
Additional explanations in the supplementary are helpful but do not leave the impression of self-contained work. Domain expertise in RL and evolutionary strategies is required to understand the paper. The paper would benefit from a brief recap of ingredients and an explanation of how TTopt plugs into the RL pipeline. The MDP formalism (B.3) does not elaborate on whether the setup is applicable in off-policy /on-policy settings, whether it makes sense to optimize the number of evaluations, and whether all requested evaluations can always be carried out.
Discussion of what entails exceeding the target reward is missing.

The paper is well-written and mainly easy to follow, except for frequent redirections to the supplement and related work for specific and essential parts of the narrative. Perhaps pivoting some text from pages 3 and 4 into the supplement won't hurt and would allow for streamlining the "story" since understanding the concepts of the tensor train decomposition does not really help to understand what is under the hood of TTopt without going into the supplement. In a sense, the matrix case (very well-written) is not representative of the multi-dimensional case due to many special handling bits.

---

> ### Author Response · Authors · 2022-08-01
> **General comments**
>
> We thank the reviewer for working thoroughly with the article and asking interesting questions. We agree that some of the points in the text may be unclear, which is an unfortunate consequence of size limitations and the cross-domain nature of the work. We added a paragraph to explain the RL aspects of the method starting at L234 in the main text and L156 in the Supplementary.

---

> ### Author Response · Authors · 2022-08-01
> **Question 1. ... algorithm can directly train quantized neural networks ...**
>
> The use of the word "quantization" indeed leads to confusion. In our experiments with TTOpt we found that treating a tensor of dimension $d$ as a higher dimensional tensor of dimension $D, D >> d$ but with a smaller mode size is very beneficial for the accuracy and speed of the method. This modification is referred to as Quantized Tensor Train (QTT), similarly to other works. At the same time, TTOpt works only with discrete parameters, which are neural network weights in RL experiments. Hence you can _only_ train quantized neural networks with TTOpt/QTTOpt. We replaced "quantized" with "discretized" in the text and emphasized this unfortunate clash of notation where it is not possible to avoid it.

---

> > ### Comment · Reviewer_uApo · 2022-08-04
> > **Response**
> >
> > > TTOpt works only with discrete parameters, which are neural network weights in RL experiments
> >
> > Unfortunately, this bit only added confusion on my side. Can the authors please elaborate on what exactly is discrete and how a neural network with discretized (using the new notation) weights is plugging into what would otherwise be replaced with a QTT? In other words, I'm trying to understand whether TTopt is applied to discretized weights of a neural network at any point, or the entire volume J is always treated as a function defined on a lattice.

---

> > > ### Author Response · Authors · 2022-08-05
> > > **Question 1. ... algorithm can directly train quantized neural networks ... (extension)**
> > >
> > > Let us consider a concrete example. Assume that we train a neural agent with $N$ weights in a given RL environment. We discretize weights, such that each weight $\theta_{i}$ ($i \in [1, 2, \ldots, N]$) can take only $K$ (let grid size be $K = 256 = 2^8$) discrete values $[ -1, -1+h, \ldots, -1 + h \cdot (K-1), 1 ]$, where $h=\frac{2}{K-1}$. The target function $J(\theta)$ is a cumulative reward the agent gets after completing one episode. As it follows, we can consider a function $J$ as an unknown $N$-dimensional tensor $\mathcal{J}$ with modes of size $K$, i.e., this tensor will have $K^{N}=256^N$ elements. We can also treat $\mathcal{J}$ as an $M$-dimensional tensor with modes of size $2$ if $2^{M} = K^{N}$ (i.e., $M = 8N$ if $K = 256$) by using QTT. At each step of the TTOpt algorithm, the following happens:
> > >
> > > - We get a new multi-index $I$ of the tensor $\mathcal{J}$ that corresponds to weights $\theta$.
> > > - We evaluate the cumulative reward $J$ with given weights $\theta$.
> > > - We update the TT-representation (in terms of the TTOpt algorithm) for the tensor $\mathcal{J}$.

---

> > > > ### Comment · Reviewer_uApo · 2022-08-08
> > > > **Overall**
> > > >
> > > > Thanks for clearing the remaining bits, I've incremented the score and confidence.

---

> > > > > ### Author Response · Authors · 2022-08-08
> > > > > **General comment**
> > > > >
> > > > > Thank you very much! We are very pleased with your high rating.

---

> ### Author Response · Authors · 2022-08-01
> **Question 2. ... why some environments benefit from finer quantization while others don't ...**
>
> Thanks for this interesting observation. The dependence on the level of weight quantization in Tab. 3 is indeed non-trivial. We can speculate that finer quantization of agent's weights leads to slower convergence of TTOpt, and hence we need to run more episodes to get the same cumulative reward. Note that we used the same number of episodes in experiments with different levels of quantization.
>
> On the other hand, neural networks with heavily quantized (low bit) weights often incur only a small drop in accuracy, see e.g. arxiv:2103.13630. The expressive power of heavily quantized policies may be enough to solve simple environments, but the convergence of TTOpt is much faster in these cases.

---

> ### Author Response · Authors · 2022-08-01
> **Question 3. What are the convergence guarantees of TTopt?**
>
> In the most general case, there are no guarantees of global/local convergence of TTOpt. The only guarantee is that the result will monotonically improve with iterations. We note that for discrete optimization problems a global optimum in the general case can not be obtained without an exhaustive search of the parameter space. However, we found that the method works well in practice and that the convergence is almost independent of the starting point if a sufficient number of steps is available. Fig. 2 in the Supplementary shows rates of convergence for 10 various benchmark functions.

---

> ### Author Response · Authors · 2022-08-01
> **Question 4. ... do the results improve further with double precision?**
>
> Thank you for pointing at this bit. This statement in the text is not entirely correct, and we changed it to "large precision". The accuracy of the final answer is determined by the curvature of the optimized function, the resolution of the discretization grid, and also by the degree of algorithm convergence. To reach convergence for very dense grids a larger number of function evaluations may be needed (please, note that we restricted the experimental budget to $10^5$ evaluations). In all experiments, we used Numpy arrays which are float32. There is no direct influence of float precision on accuracy.

---

> ### Author Response · Authors · 2022-08-01
> **Question 5. ... wouldn't it be better than the reduction procedure from LL58-61 of the supplementary ...**
>
> Thanks for pointing this out, you are right. The meaning of these lines in Algorithm A.1 is as follows. For all unfoldings of the TT-tensor we want to have "tall" submatrices, i.e., $R_{i-1} N_i \geq R_i$. This condition may not hold for several first unfoldings if $N_i$ is small and the requested $r_{max}$ is too large. In this case we reduce the rank $R_i$ to maximal possible value $R_i \gets R_{i-1} N_i$ in order to have a square submatrix at $i$-th unfolding. This condition is also enforced during the right-to-left sweep, which was missing from Algorithm A.1. We added a fix.

---

> ### Author Response · Authors · 2022-08-01
> **Question 6. ... random distribution selection ...**
>
> We used the standard Gaussian distribution $\mathcal{N}(0, 1)$ to initialize the TT-cores. We found no influence of this choice on the behavior of the algorithm. We have added a comment to Algorithm A.1.

---

> > ### Comment · Reviewer_uApo · 2022-08-04
> > **Response**
> >
> > Thanks for this confirmation. I'm curious to hear the authors' hypotheses about why distribution does not matter. I would assume that adjusting the variance of TT-cores such that elements of the full tensor have a specific variance close to that of a modeled function could positively affect the convergence.

---

> > > ### Author Response · Authors · 2022-08-05
> > > **Question 6. ... random distribution selection ... (extension)**
> > >
> > > Thank you for your inspiring question and comment. This topic needs further deep investigation which was out of the scope of this paper. We would like to say, that our extensive computer simulations indicate that performance to find global extremum was not influenced by various initial conditions.
> > >
> > > And let us explain this in more detail. According to the proposed TTOpt algorithm, we select some random TT-tensor $\mathcal{Y}_0$, which we then use to generate an initial set of column multi-indices $I_c$ (by the maxvol algorithm applied to each TT-core of $\mathcal{Y}_0$). Next, the main iteration (sweep) of the algorithm occurs by updating the set of row multi-indices $I_r$ using known column multi-indices $I_c$. Here it is important to note the following:
> > >
> > > - Instead of the initial tensor $\mathcal{Y}_0$, we could directly choose some random set of multi-indices $I_c$. For example, we can select the most distant indices from each other. As it turned out empirically, this approach does not lead to a significant improvement in the result.
> > >
> > > - As with the TT-cross method, the TTOpt algorithm quickly ``forgets'' the initial set of multi-indices if it did not match a really good initial approximation of the tensor. Accordingly, the choice of distribution for the TT-cores of $\mathcal{Y}_0$ is not particularly important (we tried to use a normal distribution with different variances, as well as a uniform distribution).
> > >
> > > - Of particular interest is the choice of some really good approximation of the target tensor as an initial approximation $\mathcal{Y}_0$. The TT-ALS (or some other) method trained on a small additional data set can be used for this. However, to date, we have not been able to achieve significant improvements by using this more complex approach.

---

> ### Author Response · Authors · 2022-08-01
> **Question 7. Why is the case of linear policies important to consider TTOpt with?**
>
> There is no substantial reason to consider linear policies. We did it because the only black-box optimized RL policies we found online were linear. We decided to try fine-tuning experiments with these weights.

---

> ### Author Response · Authors · 2022-08-01
> **Writing tips**
>
> Thank you for carefully checking the text. We implemented the proposed changes.

---

> ### Author Response · Authors · 2022-08-08
> **A note about updating the main text**
>
> In accordance with the comment of one of the reviewers, we conducted additional numerical experiments with gradient-based methods applied for all benchmark functions. We add the results to the main text (see Tab. 1 with 7 new methods/rows and explanations on LL. 198-202, 215-216, 223-226). As it turned out, for 6 benchmarks, TTOpt gives a significantly more accurate result for the same number of requests to the objective function. Compared to other methods, TTOpt is consistently accurate and avoids random failures to converge.

---

### Official Review · Reviewer_g4hT · 2022-07-11

**Rating:** 5
**Confidence:** 3
**Soundness:** 2 fair
**Presentation:** 3 good
**Contribution:** 2 fair

**Summary:**

This paper implements a novel method to find maximum elements of a tensor-shaped object function by combining quantized tensor train representation and generalized maximum matrix volume principle. The author applied this method to minimization of multidimensional functions and reinforcement learning tasks.


**Questions:**

1. The quantized Tensor train seems inconsistent with the subsection 2.4. The searching submatrix is R*R, but when the tensor is quantized to a 2-power tensor, usually R should larger than 2, thus there should be some initial step to start the algorithm.

2. As stated in line 137 and 129, each maxvol algorithm are worked in a reshaped submatrix with coloum indices randomly choosed. However, the author declare that the the algorithm will  converge to optimal row and coloum indice, which seems strange.

**Limitations:**

Comparison against bayesian optimization which is also a brunch of methods for black box optimization is missing in the paper.

**Strengths And Weaknesses:**

Strengths:
The originality is good. The algorithm combining Tensor-train and maxvol algorithm is novel and clear. The writing quality is good. The complexity is also analysed. It should be a significant work if it can work well on a wide range of tensor-shaped object function.

Weekness:
On theoretical aspect, there are no guarantee on the stability nor convergence of the algorithm.

---

> ### Author Response · Authors · 2022-08-01
> **Weakness. ... there are no guarantee on the stability nor convergence of the algorithm ...**
>
> The reviewer is correct that there are no theoretical guarantees about the convergence of the algorithm to the global minimum (we added explicit statements to the manuscript; last paragraph in Sec. "Optimization in the multidimensional case"). We note, however, that for discrete optimization problems an optimum in the general case can not be obtained without an exhaustive search of the parameter space. We found that the method works well in practice: for benchmark functions (including non-differentiable Alpine function) close to optimum solutions are found (Tab. 1 in the main text, Tab. 2-4 in Supplementary). The convergence is almost independent of the initial approximation if a sufficient number of steps is available. Fig. 2 in the Supplementary shows rates of convergence for 10 benchmark functions.

---

> > ### Comment · Reviewer_g4hT · 2022-08-09
> > **General comment**
> >
> > Thanks for the detailed rebuttal which clarifies some of my concern. I decide to increase the score.

---

> ### Author Response · Authors · 2022-08-01
> **Question 1. ... the quantized Tensor train seems inconsistent with subsection 2.4 ...**
>
> We thank the reviewer for pointing out this unclear detail. In the case of the quantized tensor train with mode size two and maximal rank $R > 2$ we simply select all indices for the first $k$ modes until $2^k > R$. We note however that this detail does not change the global behavior of the algorithm. We added an explanation to the main text (footnote 11 in Sec. 2.6).

---

> ### Author Response · Authors · 2022-08-01
> **Question 2. ... author declares that the algorithm will converge to optimal row and column indices ...**
>
> We agree with the reviewer that this claim would be too optimistic. In the text (last paragraph in Sec 2.4) we just say that we stop the algorithm if row and column indices do not change during the sweeps, which would correspond to the local optimum. Finding the local optimum does not guarantee the global optimality of the result. In practice, however, this condition is rarely met because of the small budget of iterations. We rephrased the offending paragraph to better reflect this. Also, we found almost no influence of the (random) initialization on the final result if sufficiently many sweeps are made.

---

> ### Author Response · Authors · 2022-08-01
> **Limitations. ... comparison against bayesian optimization is missing in the paper ...**
>
> Please find the comparison against Bayesian optimization in Tab. 4 in Supplementary. Compared to Bayesian methods, TTOpt is significantly faster and finds better solutions in most of our experiments.

---

> ### Author Response · Authors · 2022-08-08
> **A note about updating the main text**
>
> In accordance with the comment of one of the reviewers, we conducted additional numerical experiments with gradient-based methods applied for all benchmark functions. We add the results to the main text (see Tab. 1 with 7 new methods/rows and explanations on LL. 198-202, 215-216, 223-226). As it turned out, for 6 benchmarks, TTOpt gives a significantly more accurate result for the same number of requests to the objective function. Compared to other methods, TTOpt is consistently accurate and avoids random failures to converge.

---

> ### Author Response · Authors · 2022-08-08
> **General comment**
>
> Thank you very much for your comments, which allowed us to improve the quality of our work and clarify ambiguous points! If you have any other remarks, we will provide appropriate clarifications or make changes to the text of the paper. There is a small time for a discussion, we will be glad to fix something.

---

### Official Review · Reviewer_Aky3 · 2022-07-11

**Rating:** 6
**Confidence:** 2
**Soundness:** 2 fair
**Presentation:** 3 good
**Contribution:** 3 good

**Summary:**

The authors present a novel approach to multivariate optimisation of general objective functions making use of tensor-train decompositions over discrete state-spaces.   The TT Decomposition approximation is based on the cross-approximation method / TT-Cross of Oseledets, etc.   The main interest to use TT-cross is not for building a approximation but based on the observation that maximal elements of the tensor are highly likely to be in the maximum volume submatrix which is found using MAXVOL in TT-cross and the maximal element of
the sub-matrix increases monotonically over the iterations.    This result is hinted at, but unless I misunderstand, it is not proved or confirmed.  This is employed empirically to define a new optimisation algorithm.

The algorithm is deployed in the context of Reinforcement learning, as an alternative to evolutionary approaches, as well as some general optimisation baseline examples.  It is demonstrated that the method is competitive, and in many cases outperforms other approaches.

**Questions:**

* Why are the authors specifically considering RL as a test case for this method?  This seems to be shoe-horned in.  I don't see any necessary need for such non-differentiable optimisation methods, whereas I can think of many other situations involving high-dimensional non-differentiable objective functions.  I appreciate the motivation based on Saliman's work on Evolutionary algorithms in RL, but this seems tenuous.   Can the authors shed some light about why this is the *right* method for RL and when?  It seems that for all of the presented problems, policy gradient would've worked just fine.

* The key assumption in the proposed iterative optimisation method is that the maximum volume sub-matrix should contain the absolute maximum element, thus justifying the iteration.   However, unless I misunderstand, this isn't justified by any argument provided in this work, including the "intuition behind ttopt" presented in page 3.   I really think this needs to be studied more, because this is a crucial feature that needs to be justified.   I appreciate that this can be quite a challenge, but some intuition about why this property should be true is important.

**Limitations:**

I think the authors need to be a bit clearer about what can and cannot be guaranteed with this methodology.  It is ok if the method is heuristic but this needs to be made very explicit.

**Strengths And Weaknesses:**

Strengths:
* (Originality) This work presents an interesting approach to optimisation of (possibly non-differentiable) high-dimensional functions, exploiting tensor train decompositions, which is novel.
* (Significance) This is important as the traditional approaches to high-dimensional gradient free optimisation (e.g. BO) do not necessarily scale well with dimension, so that this offers a plausible alternative.
* (Quality) The paper is generally well written.

Weaknesses:
* (Clarity): I do not understand why the authors felt the need to bring reinforcement learning into this work.  The TTOpt algorithm is a general purpose optimisation algorithm, it is unclear why the authors felt that RL would be a unique use-case for this method, particularly given that the examples studied do not appear to involve non-differentiable loss-functions, and moreover, could be solved using policy gradient approaches.    This feels quite odd to me.
* (Significance) The key assumption in the proposed iterative optimisation method is that the maximum volume sub-matrix should contain the absolute maximum element, thus justifying the iteration.   However, unless I misunderstand, this isn't justified by any argument provided in this work.

---

> ### Author Response · Authors · 2022-08-01
> **Question 1. Why are the authors specifically considering RL as a test case for this method?**
>
> Our choice to treat RL problems with TTOpt has multiple reasons:
> - TTOpt can be directly used to optimize policies with quantized (discrete) weights which are ideal for the application in edge devices and microcontrollers. The traditional policy gradient methods are not suitable in this case without modifications.
> - The reward function in RL may not be differentiable and may pose difficulties for policy optimization, for example in the environments with stochastic reward. As the reviewer noted we considered simple deterministic environments in this initial work, but we look forward to applying the method in a harder setting.
> - Direct optimization methods often yield more stable and diverse policies compared to classical policy gradients. The application of direct search methods may be a promising alternative to traditional RL.
>
> Having noted these arguments, we stress that we tested TTOpt using a significant number of traditional optimization benchmark functions (including non-differentiable function, i.e., Alpine function; see Tab. 1 in Supplementary; for convenience, we added analytic expressions of all benchmark functions to the Table) in addition to RL benchmarks. Other applications of TTOpt in learning-based approaches may include hyperparameter tuning, latent space optimization, etc. We look forward to these perspective applications.

---

> ### Author Response · Authors · 2022-08-01
> **Question 2. ... maximum volume sub-matrix should contain the absolute maximum element ...**
>
> We agree with the Reviewer that this is a quite important property of the maximum volume principle, which needs some theoretical justification or at least some intuitive explanation. Our justification for applying the maximum volume principle in our TTOpt algorithm is based on Eq. 3 (taken from rigorously proven Theorem 1 of Goreinov et al. "How to find a good submatrix" (2010)), which guarantees that the submatrix of maximal volume will contain quite a close element to the largest element in the full matrix if the rank of a full matrix R is sufficiently low. Our very extensive numerical experiments show that the proven mathematical bound is not very tight and rather pessimistic, and in practice, the maximal-volume submatrix contains the maximum entry or at least the element which is very close to the optimal one.

---

> ### Author Response · Authors · 2022-08-01
> **Limitations. ... what can and cannot be guaranteed with this methodology ...**
>
> We agree with the Reviewer and added explicit statements in the revised version of the manuscript (last paragraph in Sec. "Optimization in the multidimensional case") that there are no rigorous guarantees of the convergence to the global minimum, nor the rate of this convergence. The only guarantee is that the result will monotonically improve with iterations.
>
> We demonstrated that our algorithm works very well in practice for a large class of difficult and widely tested  high dimensional benchmarks for global optimization (including non-differentiable Alpine function) and in most cases close to optimum solutions are found (Please see Table 1 in the main text, Tables. 2-4 in Supplementary).
>
> Our computer simulation results clearly indicate that the convergence rate is almost independent of the initial approximation if a sufficient number of steps is used. Moreover, Fig. 2 in the Supplementary shows rates of convergence behavior for 10 benchmark functions.

---

> ### Author Response · Authors · 2022-08-08
> **A note about updating the main text**
>
> In accordance with the comment of one of the reviewers, we conducted additional numerical experiments with gradient-based methods applied for all benchmark functions. We add the results to the main text (see Tab. 1 with 7 new methods/rows and explanations on LL. 198-202, 215-216, 223-226). As it turned out, for 6 benchmarks, TTOpt gives a significantly more accurate result for the same number of requests to the objective function. Compared to other methods, TTOpt is consistently accurate and avoids random failures to converge.

---

> > ### Comment · Reviewer_Aky3 · 2022-08-08
> > **Response to all comments**
> >
> > I thank the authors for their detailed rebuttals, particularly the new numerical experiments.   I agree with other reviewers that the process of discretisation of the state-space could have been better explained, particuarly for opimising weights of NNs.  However, I think the authors have addressed my concerns, so I will increase my score.

---

> > > ### Author Response · Authors · 2022-08-08
> > > **General comment**
> > >
> > > Thanks a lot! We are very pleased with your appreciation of our work. We will describe the discretization process in more detail in the text, if it will be accepted for publication.

---

### Meta-Review · Area_Chair_o4vD · 2022-08-27

**Recommendation:** Accept
**Confidence:** Certain

**Metareview:**

The basic ideas and contribution of this paper have been positively evaluated by the reviewers.
There were a few questions, but many of them have been resolved by the authors' careful replies.
There is an opinion that the comparison with reinforcement learning is inadequate, but since it is an application, this is not a major problem.


**Award:**

No

---

### Decision · Program_Chairs · 2022-09-14

Accept